# A new measurement approach for validating satellite-based above cloud aerosol optical depth

Charles K. Gatebe[1], Hiren Jethva[2,3], Ritesh Gautam[4], Rajesh Poudyal[3,5] and Tamas Várnai[3,6]

[1]NASA Ames Research Center, Moffett Field, CA, 94035, USA
5 [2]Universities Space Research Association (USRA), Columbia, MD, 21046, USA
[3]NASA Goddard Space Flight Center, Greenbelt, Maryland, 20771, USA
[4]Environmental Defense Fund, Washington, DC 20009, USA
[5]Science Systems and Applications, Inc. (SSAI), Lanham, MD 20706, USA
[6]University of Maryland, Baltimore County, Baltimore, MD 21250, USA

*Correspondence to*: Charles K. Gatebe (charles.k.gatebe@nasa.gov)

**Abstract.** The retrieval of aerosol parameters from passive satellite instruments in cloudy scenes is challenging partly because clouds and cloud-related processes may significantly modify aerosol optical depth (AOD) and particle size, a problem that is further compounded by the 3D radiative processes. Recent advances in retrieval algorithms such as the "color ratio" method 15 which utilizes the measurements at a shorter (470 nm) and a longer (860 nm) wavelength have demonstrated the simultaneous derivation of AOD and cloud optical depth (COD) for scenes where absorbing aerosols are found to overlay low-level cloud decks. This study shows simultaneous retrievals of above-cloud aerosol optical depth (ACAOD) and aerosol-corrected cloud optical depth (COD) from airborne measurements of cloud-reflected and sky radiances using the color ratio method. These airborne measurements were taken over marine stratocumulus clouds with NASA's Cloud Absorption Radiometer (CAR) 20 during SAFARI 2000 field campaign offshore of Namibia. The ACAOD is partitioned between the AOD below aircraft (AOD_cloudtop) and above aircraft AOD (AOD_sky). The results show good agreement between AOD_sky and sunphotometer measurements of the above aircraft AOD. The results also show that the use of aircraft-based sunphotometer measurements to validate satellite retrievals of the ACAOD is complicated by the lack of information on AOD below aircraft. Specifically, the CAR-retrieved AOD_cloudtop captures this "missing" aerosol layer caught between the aircraft and cloud 25 top, which is required to quantify above cloud aerosol loading and effectively validate satellite retrievals. In addition, the study finds a strong anticorrelation between the AOD_cloudtop and COD for cases where COD <10 and a weaker anticorrelation for COD >10, which may be associated with the uncertainties in the color ratio method at lower AODs and CODs. The influence of 3D radiative effects on the retrievals is examined and the results show that at cloud troughs, 3D effects increase retrieved ACAOD by about 3-11% and retrieved COD by about 25%. The results show that the color ratio 30 method has little sensitivity to 3D effects at overcast stratocumulus cloud decks. These results demonstrate a novel airborne measurement approach for assessing satellite retrievals of aerosols above clouds, thereby filling a major gap that exists in the global aerosol observations.

## 1 Introduction

The uncertainties of aerosols measurements in the vicinity of clouds has implication for the direct shortwave radiative aerosol effect and forcing on the climate system. Also, aerosols are known to exert an indirect forcing on climate by altering cloud properties and precipitation. According to the last Assessment Report of the Intergovernmental Panel on Climate Change (Boucher et al., 2013), the interactions between clouds and aerosols remain among the largest sources of uncertainty, pointing to a lack of good understanding of the aerosol-cloud system, and holding back progress in the enhancement of Earth system predictions/projections.

Space-based retrievals of aerosol optical properties in the vicinity of clouds is complex because of the difficulty in distinguishing the contributions from aerosols and clouds in top of the atmosphere (TOA) reflectance measurements. However, in the last two decades, several studies have demonstrated new approaches for aerosol retrievals in the vicinity of clouds. The absorbing aerosols such as smoke plumes, desert dust, and volcanic ash have been monitored from satellite observations in the presence of clouds using the ultraviolet measurements of Total Ozone Mapping Spectrometer (TOMS)/ Nimbus 7 (Herman et al., 1997; Torres et al., 1998), Ozone Monitoring Instrument (OMI)/Aura (Torres et al., 2012), and the Scanning Imaging Absorption Spectrometer for Atmospheric Chartography (SCIAMACHY) (De Graaf et al. 2007). The near-UV retrieval approach was extended to the visible and near-infrared spectral regions for simultaneous derivation of aerosol optical depth (AOD) and cloud optical depth (COD) based on Moderate Resolution Imaging Spectroradiometer (MODIS) measurements in regions where light-absorbing carbonaceous and dust aerosols overlay low-level clouds (c.f. Jethva et al. 2013, Sayer et al. 2016). Similarly, Waquet et al (2009) developed a method based on multiangle polarization measurements at visible and near-infrared wavelengths to retrieve aerosol properties over clouds and successfully applied it to measurements of the Polarization and Directionality of Earth Reflectances (POLDER)–Polarization and Anisotropy of Reflectances for Atmospheric Sciences Coupled with Observations from a Lidar (PARASOL) instrument. These advancements have provided hope for realizing global scale monitoring of aerosol properties over clouds, thereby filling a major gap that exists in the global aerosol observations, but significant challenges remain in the validation of above cloud aerosol products (Shinozuka et al. 2019; Redemann et al. 2020). There is no question that the above-cloud aerosol retrievals need to be validated with airborne measurements.

This study demonstrates the applicability of the color ratio method (Jethva et al., 2013; 2016), which utilizes the measurements at a shorter (470 nm) and a longer (860 nm) wavelengths for the simultaneous derivation of AOD and COD, to airborne observations. The study uses airborne data taken over marine stratocumulus clouds by the NASA's Cloud Absorption Radiometer (CAR) during SAFARI 2000 field campaign offshore of Namibia. The CAR instrument provides unique views of the cloud-aerosol system, from far, from close, or even from inside clouds–and from all the viewing directions (c.f. King et al. 1986; Gatebe et al. 2012; Gautam et al. 2016; Varnai et al. 2019; Gatebe and King 2016; Melnikova and Gatebe, 2018). The area selected has unique and reliable juxtaposition of regional and temporal patterns of meteorological conditions that are conducive to persistent low level clouds as seen from satellite imagery over the southeastern Atlantic region (cf. Figure 1); a region known to be impacted by optically thick smoke from intense biomass burning activities (agriculture crop residue burning

in central and southern Africa) (Das et al. 2020). The primary objective of this study is to retrieve aerosol optical depth above clouds using a novel airborne measurement approach of simultaneously measuring scattered radiation above and below the aircraft, and thereby demonstrate an effective observational tool to validate satellite-based aerosol retrievals above clouds.

## 2 Instruments and Methods

The Southeast Atlantic is widely used to study aerosol direct and indirect radiative effects because of the presence of stratiform marine clouds over ocean and the annual recurrence of very high concentrations of biomass burning aerosols between June and September (cf. Das et al. 2020; De Graaf et al. 2007; De Graaf et al. 2012; Keil and Haywood 2003; Meyer et al. 2013; Sayer et al., 2016; Pistone et al. 2019; LeBlanc et al. 2020). The measurements analyzed here were taken aboard the University of Washington's Convair-580 research aircraft. During several portions of the flight analyzed here, the aircraft followed a

circular flight track (Figure 1) at a near-constant distance from the cloud top (~650 m) occurring below ~1 km altitude (Gatebe et al 2003; Sinha et al. 2003).  The image acquired by MODIS/Terra on the same day at about 09:25 UTC (see Figure 1, map inset), shows widespread clouds over the entire Namibian coast. There were reports during the Southern African Regional Science Initiative's (SAFARI 2000) dry season campaign (Swap et al., 2002) that optically thick smoke that originated from intense biomass burning activities was advected over to the marine stratiform clouds off Namibian coast. The CV-580 flight

began just prior to 10:00 UTC and ended at about 13:00 UTC. Table 1 summarizes the times and locations of the cases analysed, which are labelled alphabetically, a-p, based on the time of observations.

### 2.1 Aircraft & Sensors

The CAR instrument flew aboard the UW CV-580 research aircraft (Figure 2a), and obtained the bidirectional reflectance distribution function (BRDF) over an extensive and persistent stratocumulus cloud deck with an overlaying smoke aerosol

layer.  The aircraft was also equipped with other instruments to measure gases, aerosols and radiation (see Appendix A by P. V. Hobbs in the work of Sinha et al. 2003).  Figure 2b shows a cutaway drawing of CAR. The instrument is approximately 72 cm long, 41 cm wide, and 39 cm deep and weighs 42 kg. CAR was designed primarily to image the sky and surface at an instantaneous field of view (IFOV) of 1° through a 190° wide plane as shown in Figure 2c.  CAR measures both transmitted and reflected radiances at 14 narrow spectral bands located in the ultraviolet, visible and near-infrared (0.340-2.303 μm; Figure

2d).  This combination provides a convenient and efficient means of obtaining complete BRDFs for any surface type at a landscape level and ensures that surface albedo, which is an angular-weighted integration of the reflection function over a hemisphere, can be derived from these measurements covering the required angular range (Nicodemus et al., 1977; Kimes et al., 1987).

During the BRDF measurements over the marine stratiform clouds, the instrument obtained unique views of the

cloud-aerosol system, scanning from zenith to the horizon and then from the horizon to nadir, and covering the entire 360° range of azimuthal directions as the aircraft flew in a circular flight track (c.f. Gatebe et al. 2003; Fig. 3). The quicklook RGB

image in Figure 3 (R=1.04 μm, G=0.87 μm, and B=0.47 μm) illustrates measurements taken from 12:27 UTC to 12:54 UTC. The sun can be seen in the sky at about 33° view zenith angle, which also corresponds to the solar zenith angle, and a bright cloud system is seen on the image from view zenith angles 90-180°. The horizon coincides with the 90° view zenith angle, which is easily identified by the contrast between the sky and surface. In this image, the principal plane is defined by the vertical plane containing the sun and the plane that is equidistant between two solar disks.

Note that the circular flight track during the BRDF measurements above the clouds (~650 m) is about 4 km in diameter, and with an aircraft bank angle of 20-30°, which is compensated by CAR to help maintain the full 180° view from zenith to nadir, the plane took ~3 minutes to complete an orbit. The marine stratiform clouds are generally characterized by a well-defined cloud top height corresponding to a strong boundary layer inversion. Given this viewing geometry of the cloud-aerosol system, the CAR measurements permit the retrieval of aerosol optical properties above clouds separated into above and below the aircraft, plus the cloud optical properties, using the color ratio method. These measurements provide the best data for validating above cloud aerosol retrieved from satellite measurements, analogous to the validation of cloud-free aerosol retrievals from satellites, which is typically done with observations from the AErosol RObotic NETwork (AERONET) ground-based sunphotometer network (Holben et al. 1998).

## 2.2 The color ratio method and its application to airborne observations

The color ratio (CR) method has been used to simultaneously retrieve the above-cloud aerosol optical depth (ACAOD) and aerosol-corrected COD from OMI (Torres et al. 2012) and MODIS observations (Jethva et al. 2013; 2016). The technique is physically based on the reduction of the ultraviolet (UV), visible (VIS), and near-infrared (NIR) radiation reaching the top of atmosphere, due to particle absorption above cloud. The effects of aerosol absorption have a spectral signature, in which the absorption strength is found to be stronger at shorter wavelengths than at longer wavelengths. This produces a strong color effect in spectral measurements, hence, the name color ratio method. The method employs the VLIDORT V2.6 polarized radiative transfer model (Spurr, 2006) for the simulation of LUT reflectances. VLIDORT treats the outgoing radiance in a pseudo-spherical geometry. Therefore, it is expected that the aerosol radiance simulation at slant geometry, i.e., viewing zenith angle > 70° may not carry the same accuracy as the case with lower viewing angles. This may result in less accurate retrievals at extreme viewing geometries. Additionally, larger retrieval errors at lower cloud optical depth measurements and heterogeneity in aerosol and cloud fields also add to the apparent dependence on scattering angle.

The aerosol microphysical-optical properties of carbonaceous smoke model and radiative transfer configurations assumed in the radiative transfer simulations are shown in Table 2. The aerosol model used here in the ACAOD inversion is identical to the one employed in Jethva et al. (2016), where the MODIS retrievals of ACAOD were found to be in very good agreement (RMSE~0.05 and 99% matchups within predicted uncertainty) against those directly measured from AATS sunphotometer. The results implied that the aerosol microphysical-optical properties assumed in the inversion based on the long-term, ground-based AERONET inversion at an inland site Mongu, are suitable for ACAOD retrievals over the adjacent Atlantic Ocean. The

retrieved ACAOD at 470/860 nm is converted to its value at 500 nm according to the spectral extinction assumed in the selected aerosol models.

        The near-UV based color ratio algorithm has been applied to the long-term record of OMI to derive a global product of ACAOD (Jethva et al., 2018). The ACAOD product has been validated against airborne measurements taken from HSRL-2 lidar operated during the ORACLES campaign conducted over the south eastern Atlantic Ocean. On the other hand, the

ACAOD derived from the visible/near-IR observations of MODIS was validated against the direct AOD measurements acquired from airborne NASA Ames Airborne Tracking Sun Photometer (AATS) and Spectrometer for Sky-Scanning, Sun-Tracking Atmospheric Research (4STAR) sunphotometers operated during different field campaigns (Jethva et al. 2016). In both OMI and MODIS validation studies, the satellite-retrieved ACAOD product was found to agree well with the airborne measurements within the expected uncertainty limits associated with the inversion technique, which mainly arises from the

chosen aerosol model and its absorption properties.

        Here, the CR method was applied to CAR observations, which include direct and diffuse solar radiance (or sky radiance), at eight spectral channels (see Fig. 4.). The direct solar component is given by the extra-terrestrial solar radiance attenuated by atmospheric absorption and scattering. On the other hand, sky radiance results from single and multiple scattering processes due to interaction of sunlight with aerosols and gas molecules. Atmospheric gas molecules (e.g. nitrogen, oxygen,

carbon dioxide, ozone, water vapor, etc.) and aerosols are likely to strongly affect the solar radiance in the visible and near infrared regions. The attenuation (scattering and/or absorption) by each atmospheric constituent is strongly dependent on wavelength and can be determined through the optical thickness using simple parametric models (e.g. Zibordi and Voss, 1989). In the case of CAR measurements close to the sun (solar aureole), the signal from the direct solar radiance measurements saturate the detectors and therefore pixels that are especially close to the solar direction (scattering angles are ≤10°) should be

excluded from any retrieval (Gatebe et al. 2010). The sky radiance distribution seen here is typical of clear skies (cloud free), where the radiance of a point in the sky depends both on its position relative to the sun (i.e., azimuth angle) and on its airmass number (i.e., zenith angle).  The sky radiance distribution is generally symmetrical about the principal plane, where the maximum value of the sky radiance for each wavelength is observed. This is illustrated in Fig. 4e:  45° ≈ 315°; 90° ≈ 270°; 135° ≈ 225° for λ >0.4 μm. The minimum values of sky radiance are found to be in the area directly opposite to the sun's

position.

        The CAR observations are indicative of the presence of absorbing aerosols above the clouds due to apparent brightening/darkening, which is evident when looking at the measured sky radiances /cloud bidirectional reflectance factor (BRF) (cf. Fig. 4). Aerosol loading has a strong influence, especially in the forward scattering directions (relative azimuth angle (φ) < 90° and φ>270°), with reflectances in the shorter wavelengths (e.g. 0.38 μm) larger by a factor of >2 relative to

the longer wavelengths (e.g. 1.22 μm; Fig. 4e). The asymmetry depicted in Fig. 4e is largely attributed to aerosol scattering and not to Rayleigh scattering, as the latter is expected to exhibit symmetrical distribution in either scattering directions. More interestingly, there seems to be a strong aerosol absorption signal above clouds. It is well known that clouds reflect uniformly across the visible-near-IR spectrum, however, the presence of absorbing aerosols above clouds (in this case smoke transported

from southwestern Africa) induces an overall absorption or darkening in the UV and shorter visible wavelengths, thus resulting in a strong reflectance gradient from UV to blue to near-IR spectrum, ~35% reduced reflectance at 0.34 µm compared to that at 1.04 µm, as seen in Fig. 4f. Overall, the positive spectral gradient seen in Fig. 4f, is normally associated with cloud darkening at the shorter wavelengths (cf. Gautam et al. 2016).

## 2.3 The 3D radiative transfer simulations

To examine 3D influences in CAR retrievals, we performed 1D and 3D radiative transfer simulations using the Monte Carlo model that powers the online simulator of 3D radiative processes that was created as part of the I3RC (Intercomparison of 3D Radiation Codes) project and is publicly available at http://i3rcsimulator.umbc.edu/. This model was validated through I3RC intercomparison experiments (e.g., Cahalan et al., 2005) and was used in several other studies (e.g., Várnai et al., 2013). The key simulation parameters are listed in Table 3; additional details and the results of the simulations are discussed in Section 3.4.

## 3 Results

### 3.1 The observations

Figures 5 and 6 show the full BRF of low stratiform clouds at selected wavelengths of 0.472 µm and 0.870 µm, respectively, from each of the 16 different cases described in Section 2. The two wavelengths form the basis of the "color ratio" method for the simultaneous retrieval of above-cloud aerosol optical depth (ACAOD) and cloud optical depth (COD). The spectral BRF of stratiform clouds observed in the 16 cases is highly anisotropic due to a combination of factors ranging from cloud heterogeneity (including sub-pixel heterogeneity), solar illumination geometry, sensor viewing geometry, and cloud parameters such as optical thickness and effective radius (cf. Cornet et al. 2018). The 16 cases have a range of solar zenith angles (23°< SZA <36°). Measurements span an area of ~55 km (N-S) × ~12 km (E-W), with most cases (9 cases: cases h-p) concentrated over a much smaller area (~8 km x ~4 km) (cf. Fig. 1). The observations were taken at approximately the same altitude (Table 1: cases a-d: 1420 -1541 m above mean sea level (AMSL) & cases h-p: 1608-1616 m AMSL), implying that corresponding pixels for different cases have similar measurement scale. The only exceptions (cases e-f) were taken at different altitudes during the aircraft spiral from 1814 m to 3369 m AMSL. The cloud top height was ~1000 m AMSL (Sinha et al. 2003) and the cloud geometrical thickness was at most 300 m (cf. Melnikova and Gatebe 2019, subsection 2.2). Based on these characteristics, the 16 cases may be grouped into three groups (cf. Table 1): Group 1: cases a-d (SZA≈24°, measurements were taken close to each other in time, Δt <16 minutes, altitude ≈1508 m and the location is about the same as shown in Figure 1). Group 2: cases e-g (SZA≈24°, Δt <6 minutes, altitude ≈ variable from low to high, and same location near the Namibian coastline as shown in Figure 1. Group 3 cases h-p (SZA≈34°, Δt <23 minutes, altitude ≈1614 m, and the location is about the same as shown in Figure 1). Since the stratiform clouds are formed and maintained by a balance of various marine boundary

layer processes (cf. Duynkerke and Teixeira 2001; Woods 2012; Feingold et al. 2017), the variations in the BRF patterns with time, especially where other parameters are similar, are possibly linked to formation of open cells caused by the drizzle-cloud dynamical interactions and inevitably leading to changes in the cloud liquid water path and BRF. The pronounced circular brightness feature (see cases h-p, Fig. 5, $\lambda$ = 0.470 μm, or Fig. 6, $\lambda$ = 0.870 μm) shows a cloud bow (or primary rainbow), which is typical of water clouds (cf. Gatebe et al. 2003, where case h was analysed in detail). Figure 7 shows the derived

spectral albedo (with atmosphere) for all the 16 cases at $\lambda$=0.470 μm and $\lambda$= 0.870 μm (see Table 4 for the spectral albedo (with atmosphere) for all the wavelengths). Clearly, Group 3 cases had higher spectral albedo and was optically thicker, while Group 2 cases from near the Namibian coastline had the lowest spectral albedo (with atmosphere). It is interesting to note that the spectral albedo remains almost constant in Group 2 cases, despite the change in measurement scale during the spiral. In the following subsections, we will examine how the surface reflectance anisotropy impacts the retrievals of the optical depth

(both clouds and aerosols) using the color ratio method.

### 3.2 The retrieved ACAOD and COD

Figure 8 shows the retrieved AOD for aerosol layers located above the aircraft-level  (AOD_sky)) derived from the observed diffuse sky radiance by CAR.  The retrievals were performed using a single-channel fit at 470 nm between the observed sky radiance aerosol look-up table accounting for the variations in AOD and geometry. Note that the aerosol model

used for AOD_sky retrievals was the same for the inversion of AOD below aircraft (AOD_cloudtop). It is complicated to characterize and model the anisotropic effects of reflecting clouds with varying optical depths on the hemispherical diffuse sky radiances measured by CAR. Therefore, we adopted a simple approach to account for these effects, at least partially, by retrieving AOD above the aircraft assuming an averaged underneath cloud optical depth field retrieved from the AOD_cloudtop inversion for each CAR BRDF case. For the most part the hemispherical distribution of retrieved AOD_sky

along the azimuth direction is found to be smooth and near-uniform suggesting that the sky retrievals of AOD aren't significantly affected by the cloud anisotropy and that the simple approach of assuming an averaged value of COD for the full azimuthal scan works reasonably well in capturing the cloud effects on the sky radiances.  The angular pattern in cases a-d is similar and in good agreement with the airborne direct sunphotometer measurements as discussed later (Figure 12 and Table 1).

The retrieved AOD below the aircraft (AOD_cloudtop) for all the 16 CAR BRDF cases are shown in Figure 9. The white areas in each polar plot are devoid of AOD_cloudtop retrievals either due to no cloud detection and/or the observations fall outside the color ratio vs. reflectance look-up table domain including extreme viewing geometry. In almost all cases (a-p),

the retrieved AOD_cloudtop shows a dependence on viewing zenith angle, where lower (higher) AOD_cloudtop values are associated with slant (near-nadir) viewing angles (see also Figure 11 – scatter plots of AOD_cloudtop vs COD). Such gradient in the retrieved AOD_cloudtop can result from the limitations of the radiative transfer calculations at slant angles and the fact that CAR observations are interpreted within the look-up table after linearly interpolating between aerosol geometry nodes. The nodes in geometry used in the RT calculations include solar view zenith angles (sza_nodal), view zenith angles (vza_nodal), and relative azimuthal angles (raa_nodal) (see Table 2). Another salient feature of the retrieved AOD_cloudtop field is the intermittent patches of high AODs that extend in the viewing zenith direction along an azimuthal plane. A careful qualitative inspection of this feature with BRFs measured at 0.47 μm (see Fig. 5) and 0.87 μm (Fig. 6) reveals that the higher AODs are spatially collocated with relatively lower values of BRF, indicating that these observations belonged to either clear-sky or partially cloudy sky or thin heterogeneous scenes, for which the assumption of fully overcast thick homogeneous pixels made in the CR algorithm breaks down. Under such situations, it is expected that the uncertainty in the retrieved AOD_cloudtop would be larger than the expected errors due to other algorithmic assumptions. This issue is explored further in subsection 3.4 under the influence of 3D effects on the retrieved AOD_cloudtop and COD.

Another important observation in Fig. 9 is the increasing magnitudes of AOD above cloud for the cases e, f, and g. Table 1 shows that the altitude of aircraft for these three cases was recorded as 1533±2, 1814±259, 2646±223 meters above mean sea level. It is expected that as the aircraft altitudes moves higher in the atmosphere, the CAR sensor would see an aerosol layer of greater geometrical thickness, thereby resulting in greater aerosol extinction and AOD. The retrieved AOD_cloudtop for these cases precisely demonstrates this effect by showing increasing magnitudes for higher aircraft altitudes.

The color ratio algorithm, along with the above-cloud AOD, also co-retrieves aerosol-corrected cloud optical depth, which is shown in Figure 10. Unlike aerosol fields, both seen above and below the aircraft level show more homogeneous distributions, the cloud optical depth fields retrieved from most of the cases show a great deal of variability along the azimuthal plane. Except for the cases m, n, o, and p, all other cases (a through l) show overall higher cloud optical depth in the back scattering directions shown in the bottom hemisphere opposite to the Sun and between the azimuth angle 90° and 270°. Unlike polar orbiting satellite observations at a fixed geometry for a given overpass, the CAR measurements offer a complete picture

over all the viewing directions relative to the sun direction. This unique observational geometry provides increased information content that would allow quantification of the effects of angular reflectance distribution in remote sensing retrieval algorithms.

### 3.3 The relationship between AOD_cloudtop and COD

Figure 11 shows the scatter plots of AOD_cloudtop vs COD for view zenith angles 0°-30° (blue color), 30°-60° (green color), and 60°-90° (red color), which shows very interesting patterns. The retrievals of AOD_cloudtop are found to exhibit a systematic dependence on COD (similar to an exponential decay function), especially the blue color and green color dots, where larger values of AOD_cloudtop correspond to lower values of COD and crawling along the x-axis on the right as COD increases. An exception to this rule are the retrievals made at the higher view zenith angles, 60°-90° (red color), where the retrieved ACAOD remains low (<0.2) despite an increase in the COD, which seems unrealistic and confirms some of the limitations of the color ratio method. Another exception is seen in cases e, f and g, where AOD_cloudtop vs COD show no clear dependence on viewing zenith angle and COD was around 5, indicating that these observations belonged to either clear-sky or partially cloudy sky or thin heterogeneous scenes, for which the assumption of fully overcast thick homogeneous pixels made in the CR algorithm breaks down. The relationship between the two retrieved quantities appears to be confined for COD<10, after which both retrievals are found to be not related to each other. Such observed dependence was expected as noticed in the color ratio algorithm introduced in Jethva et al. (2013). The uncertainties in satellite ACAOD inversion is known to be larger at lower CODs. This is because the retrieval domain space, i.e., color ratio versus reflectance at a longer wavelength, at lower CODs becomes narrower with steep changes in the color ratio, especially at COD<10. Therefore, any uncertainty in the assumptions made in the retrieval algorithm, i.e., single-scattering albedo, an assumption of fully overcast pixels, and linear interpolation between the nodes where reflectances and its ratio of a joint aerosol-cloud scene behaves non-linearly would result in the amplification of the error in the retrieved ACAOD. These artifacts are more pronounced at lower values of both ACAOD and COD, where uncertainties in the retrieved ACAOD could reach 40% to 80% at COD<10 and ACAOD<0.5 typically observed in the present CAR AOD retrievals (Jethva et al., 2013, Table II). Figure 11 results also suggest a strong inverse relationship between the AOD_cloudtop and COD for cases where COD <10, and a weaker inverse relationship for COD >10. Additionally, studies (e.g. Torres et al., 2012; Jethva et al., 2018) estimated uncertainty limits in ACAOD for typical range of satellite-viewing geometry (i.e., solar zenith angle 20-40°, viewing zenith angle 0-40°, and relative azimuth angle 100-150°), while varying the single-scattering albedo and aerosol layer height. The error estimates of

ACAOD, not reported in these papers though, were found to be near-stable as a function of geometry in the stated ranges. A near-uniform retrieval of sky-looking AOD (above-aircraft and clouds) shown for different CAR profiles in Figure 8 further demonstrates the stability of the algorithm for viewing zenith range 0-60°. At slant angles >60° and around the edge of the scan, the limitation of radiative transfer calculations due to its pseudo-spherical treatment in the RT code restricts the accuracy of AOD inversion. However, we note that no explicit cloud-screening was performed on the measurements. All measurements go through the ACA algorithm where if they fit into the retrieval domain, i.e., color ratio vs. reflectance 860 nm, then a corresponding retrieval of ACAOD and aerosol-corrected COD are obtained. It is possible that heterogeneity in aerosol and cloud fields in the observed scene can introduce uncertainty in the retrievals. For instance, a mixture of cloudy and cloud-free scenes observed in a particular measurements can affect both AOD and COD inversions.

Figure 12 shows the two main aerosol-above-cloud retrieved parameters, namely AOD_sky, when CAR views upward flying above the cloud field, and the AOD below aircraft (AOD_cloudtop), when CAR views downward measuring the cloud field averaged over all the viewing directions (see also, Table 1, columns 6-9). The summation of AOD_sky and AOD_cloudtop provides the column AOD above the stratocumulus cloud fields (ACAOD), as retrieved from CAR measurements over marine stratus clouds during SAFARI 2000 in the southeast Atlantic region. In addition to the two aerosol-above-cloud parameters retrieved from CAR, Fig. 12 also shows simultaneous COD retrievals using CAR measurements as well as AOD retrievals from the AATS sunphotometer that made coincident measurements of AOD on the UW CV-580 flights. The AOD retrievals from AATS are based on direct Sunphotometer measurements and therefore represent aerosol loading above the aircraft-level.

In the case of flight transects shown in Fig. 1, the AATS AOD retrievals were largely obtained above the marine stratocumulus clouds. However, when the cloud top is well separated from the aircraft, i.e., the altitude of aircraft is higher than that of the cloud tops; the AATS measurements do not capture the aerosol layer below the aircraft as the instrument is always pointing upwards, toward the Sun. Therefore, the reported AOD data from AATS is not representative of the total column AOD above clouds, unless the aircraft is flying at the same altitude where cloud top is located. Often, the altitude difference is not negligible, for example, during the SAFARI flights shown in Fig. 3, there was a clear separation of ~600 m between the aircraft and cloud top. Specifically, the CAR-retrieved AOD_cloudtop captures this "missing" aerosol layer caught

between the aircraft and cloud top, which is in addition to the AOD_sky retrieved above the aircraft level. The latter quantity is equivalent to that retrieved by AATS, whereas AOD_cloudtop is the remainder of the column AOD that we retrieve from CAR in this study. For these reasons, Jethva et al. (2016) in validating MODIS-retrieved ACAOD for the same September 13, 2000 AATS flight extrapolated the airborne measurements from the respective altitudes to cloud-top using a detailed profile measurements and associated altitude-AOD polynomial in order to make the comparisons between satellite and airborne measurements consistent.

To illustrate the various retrievals, we consider flight measurements from cases h-p. The COD associated with the marine stratocumulus clouds (cases h-m) vary between 15 and 20 (Fig. 12). These retrievals (for cases h-m) are based on relatively homogeneous clouds observed during the three separate circular measurements obtained from transects a-d, e-g and h-p. These relatively homogeneous and similar sets of circular transects are also noted in the BRF polar plots shown in Fig. 6h-m. The simultaneous retrievals of Sky_AOD show moderately high aerosol loading, AOD = 0.5 across circles h-m, which is in very close agreement with the AATS_AOD retrievals. The consistency in AOD retrievals (above the aircraft level) between the two disparate measurement approaches, i.e. AATS and CAR, is generally found throughout the data obtained from the 16 cases (a-p), as indicated by the high correlation ($R^2 = 0.92$) between the two retrievals shown in Table 1. However, the central distinction here is that the CAR approach also allows us to directly retrieve aerosols above clouds that are present below the aircraft level (AOD_cloudtop). For instance, in cases h, the AOD_cloudtop is 0.18 and the Sky_AOD is 0.50, implying the total above-cloud column AOD is 0.68 or 31% higher relative to the AATS_AOD retrieval. Overall, we find AOD_cloudtop ranging between 0.18 and 0.41 from the 16 cases shown in Fig. 12, indicating a notable enhancement of the overall presence of aerosols above clouds. These observations show that a significant aerosol layer is not captured by the aircraft sunphotometer, indicating the strength and effectiveness of near-simultaneous multiangular measurements scanning the sky and surface, as demonstrated in this study using CAR measurements.

### 3.4 The influence of 3D effects on the retrieved ACAOD and COD

Numerous earlier studies indicate that passive remote sensing of both cloud and aerosol properties can be significantly impacted by three-dimensional (3D) radiative processes (e.g., Marshak and Davis, 2005; Wen et al., 2006; http://i3rc.gsfc.nasa.gov/Publications.htm). Since the impact of 3D effects is different for different observations and retrieval algorithms (e.g., Cornet et al., 2018), we next examine the impact of 3D effects on the CAR aerosol and cloud retrievals

discussed above. Our goal does not lie in providing quantitative estimates of 3D effects; instead we examine whether 3D

effects are likely to play a substantial role in shaping the behavior of CAR-retrieved cloud and aerosol optical depths.

Our tests consider the scene shown in Figures 5k, 6k, 9k and 10k as a representative of heterogeneous areas with potentially significant 3D effects. The figures show that around 60° azimuth angle, CAR observed a roughly 300 m wide and very long trough in which the retrieved COD drops by roughly 50% (Fig. 10k), while the retrieved AOD_cloudtop increases by roughly 50% (Fig. 9k). Figures 9, 10, and 11 show that this behaviour is not unique, and that in many cases with COD values below

10 or sometimes even 20, the retrieved AOD values increase sharply as COD decreases. In principle, this behaviour appears consistent with earlier findings that showed 3D effects to increase retrieved AOD values for pixels that were surrounded by brighter (thick-cloud-covered) areas (e.g., Wen et al., 2013).

As discussed in Section 2.3, we examined the impact of 3D radiative effects through Monte Carlo simulations whose results are listed in Table 5. In each row of this table, the left column indicates whether or not below-CAR aerosols (BCA)

were considered, what the cloud optical depth was at the trough center, and whether the simulations considered 1D or 3D radiative processes. The indicated uncertainties come from Monte Carlo simulation noise.

Since COD retrievals are shaped mainly by the 0.87 µm reflectance values, 3D BRFs exceeding 1D BRFs by about 25% for COD=7 indicates that 3D radiative processes significantly enhance CAR BRFs and thus the COD values retrieved in the center of the trough—which means that 3D effects make the COD drop in the trough appear less deep than it really is. This

behavior is consistent with earlier studies showing that radiative smoothing (caused by the diffusion of photons scattered from thick to thin areas) make horizontal cloud variability appear less strong than it really is. Several studies proposed counteracting this effect by artificially roughening the retrieved COD fields (e.g., Marshak et al., 1998; Zinner et al., 2006), but these methods are yet to gain wide usage. By performing additional simulations, we found that if we decreased COD at the center of the trough from 7 to 4.7, 3D simulations would yield 0.87 µm BRF values around 0.32—thus resulting in hypothetical retrievals

yielding COD=7 (similar to the actual CAR retrievals). We note, however, that the value of 4.7 depends on our assumption of cloud base altitude (hence cloud geometrical thickness), and so it is somewhat uncertain.

Regarding aerosol retrievals, we first examine how 3D radiative processes affect the key signal of our ACAOD retrievals, which is the impact of below-CAR aerosols (BCAs) on the BRF(0.47 µm) / BRF(0.87 µm) color ratio (CR) values. Specifically,

we compare the CR values for the BCA and noBCA cases, and check whether the CR-difference is similar in 1D and 3D radiative simulations:

$$((CR3D(BCA) - CR1D(no\ BCA)) / (CR1D(BCA) - CR1D(no\ BCA)) = 1.052 \pm 0.02$$

While the calculations above used the retrieved value of COD = 7 at the center of the linear trough, we also tested whether the results change if the 3D simulations use COD=4.7 instead:

$$((CR3D, COD=4.7(BCA) - CR1D(no\ BCA)) / (CR1D(BCA) - CR1D(no\ BCA)) = 1.075 \pm 0.02$$

These results indicate that 3D processes strengthen the impact of BCAs on CR values by about 3-10%.

To estimate the impact of these CR changes on retrieved ACAOD values, we examined the non-linearity of the CR-ACAOD relationship using additional 1D Monte Carlo simulations. These simulations used the same setup as in Table 2, except that below-aircraft ACAOD values were increased by 20%. The simulations (identified by the subscript IBCA) gave $BRF_{IBCA}(0.47\ \mu m) = 0.24523 \pm 0.00004$ and $BRF_{IBCA}(0.87\ \mu m) = 0.32069 \pm 0.00006$, yielding $CR_{IBCA} = 0.76469 \pm 0.00027$. Comparing the impact of original and increased BCA amounts on CR gives

$$(CR_{IBCA} - CR_{noBCA}) / (CR_{BCA} - CR_{noBCA}) = 1.1900 \pm 0.0089.$$

This indicates that a 20% enhancement in ACAOD causes a 19% enhancement in the CR signal, which implies that a 10% change in CR is consistent with a 10% * 20 / 19 = 10.5% change in ACAOD. Considering the uncertainties, we can say that the 3-10% impact of 3D effects on CR values corresponds to a 3-11% impact on retrieved ACAOD values.

To understand this result, we need to consider both the radiative smoothing discussed above for COD retrievals, and the 3D process often called "bluing" (e.g., Marshak et al., 2008). Bluing occurs when nearby thick clouds reflect more sunlight than the clouds in the field-of-view do, and some of the extra reflection is then scattered into the instrument field-of-view by air molecules and aerosol particles that reside between the cloud and the sensor. As expected, Table 5 reveals that 3D processes do indeed enhance BRFs: For COD=7, $BRF_{3D}$ values exceed the corresponding $BRF_{1D}$ values at both 0.47 $\mu m$ and 0.87 $\mu m$. However, the table also reveals that given a certain 0.87 $\mu m$ BRF value, 3D and 1D processes yield fairly similar 0.47 $\mu m$ BRFs and thus color ratios: $BRF_{0.47\ \mu m,\ COD=4.7,3D} \approx BRF_{0.47\ \mu m,\ COD=7,1D}$ and $CR_{3D,\ COD=4.7} \approx CR_{1D,\ COD=7}$.

The weak impact of 3D effects on CR is likely due to two factors. First, while the bluing process implies a larger molecular and aerosol scattering enhancement at 0.47 $\mu m$ than at 0.87 $\mu m$ (i.e., a higher CR), this is partially compensated by the aerosol

absorption cross section being larger at 0.47 µm than at 0.87 µm. Second, much of the 3D effects that cause the enhancements

apparent in Table 5 are likely caused by the in-cloud radiative smoothing process discussed above, which causes similar relative

enhancements in the trough BRFs at 0.47 µm and 0.87 µm: Cloud droplets, which cause radiative smoothing through multiple

scattering, have similar scattering properties at 0.47 µm and 0.87 µm.

We note that simulations (not shown) indicate that 3D effects would have similar or even weaker influence on ACAOD

retrievals over the linear trough if the measurements were taken not by CAR flying only 600 m above the clouds, but by a

380 satellite passing overhead. This is because the compensating effect of aerosol scattering and absorption and the spectrally

neutral in-cloud radiative smoothing cause 3D relative enhancements that are spectrally quite neutral.

Overall, the results discussed above imply that 3D radiative processes had a significant impact on retrieved cloud optical

depths, but also that the 3D impacts on retrieved ACAOD values is fairly small and is not the main reason for the retrieved

ACAOD values increasing over thin clouds.

**4. Conclusion**

In conclusion, the study accomplished the simultaneous retrieval of above cloud total aerosol optical depth (ACAOD) and

aerosol-corrected Cloud Optical Depth (COD) from airborne CAR measurements of cloud-reflected and sky radiances using

the color ratio method. The ACAOD is partitioned between the AOD below the aircraft (AOD_cloudtop) and the AOD above

the aircraft (AOD_sky) with full angular coverage provided by the CAR measurements. The study demonstrates a novel

measurement approach for retrieving and quantifying aerosols above clouds, in particular recovering the aerosol layer between

cloud tops and aircraft level that is missed in typical airborne sunphotometer measurements of above-cloud-aerosols. Overall,

this work provides a path forward for filling a critical gap in aircraft-based sunphotometer measurement strategies that are

currently used to validate satellite retrievals of the ACAOD.

The results show a strong anticorrelation between the AOD_cloudtop and COD for cases where COD <10, and a

395 weaker anticorrelation for COD >10. The impact of 3D radiative effects on the retrievals is examined and the results show

that at cloud troughs, 3D effects increase retrieved ACAOD by about 3-11% and retrieved COD by about 25%. This indicates

that the color ratio method has little sensitivity to 3D effects at overcast stratocumulus cloud decks. The results also display

good agreement between CAR and sunphotometer measurements of the above aircraft AOD. However, the results also show

that the use of aircraft-based sunphotometer measurements to validate satellite retrievals of the ACAOD is complicated by the

400 lack of information on AOD below the aircraft, indicating the strength and effectiveness of near-simultaneous multiangular

measurements scanning the sky and surface, as demonstrated in this study using CAR measurements.

**Acknowledgements**

We are grateful to all our colleagues who helped in many ways and made it possible to collect the analyzed observations. This research was primarily funded by NASA's Atmospheric Composition Campaign Data Analysis and Modeling solicitation, ACCDAM, which is managed by Hal Maring. CAR SAFARI 2000 data acquisition was funded by the EOS Project Science Office led by Michael King.

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

Table 1: Retrieved parameters from a total of 16 CAR Bidirectional Reflectance-distribution Function (BRDF) cases taken on September 13, 2000 during SAFARI-2000 campaign. AOD values are derived at λ= 0.500 μm.

| Case | Location (°S, °E) | Time (UTC) HH: MM: SS | Solar Zenith (°) | Mean Aircraft Alt. , m, (AMSL) | Retrieved COD | Retrieved AOD_cloudtop | Retrieved AOD_sky | AATS_AOD |
|------|-------------------|------------------------|-------------------|---------------------------------|----------------|-------------------------|--------------------|-----------|
| a | 20.67, 13.13 | 10:44:51 | (24.67) 24.36 | 1420±40 | 12±5 | 0.25±0.14 | 0.56±0.09 | 0.56±0.03 |
| b | 20.62, 13.12 | 10:50:47 | 24.11 | 1540±2 | 12±5 | 0.24±0.16 | 0.55±0.09 | 0.55±0.03 |
| c | 20.62, 13.12 | 10:53:21 | 24.04 | 1541±2 | 11±5 | 0.27±0.18 | 0.55±0.09 | 0.55±0.03 |
| d | 20.61, 13.13 | 11:01:13 | 23.95 | 1533±2 | 8±3 | 0.34±0.18 | 0.58±0.09 | 0.55±0.03 |
| e | 20.24, 13.20 | 11:18:00 | 23.94 | 1814±259 | 7±2 | 0.32±0.21 | 0.55±0.10 | 0.55±0.03 |
| f | 20.24, 13.20 | 11:21:00 | 24.09 | 2646±223 | 7±3 | 0.33±0.19 | 0.45±0.11 | 0.48±0.03 |
| g | 20.25, 13.20 | 11:23:47 | (24.25) | 3369±250 | 7±4 | 0.41±0.17 | 0.32±0.14 | 0.40±0.03 |
| h | 20.26, 13.22 | 12:28:07 | (31.70) 31.88 | 1608±3 | 19±7 | 0.18±0.10 | 0.50±0.11 | 0.52±0.03 |
| i | 20.48, 13.10 | 12:30:34 | (32.10) 32.28 | 1613±2 | 19±7 | 0.19±0.10 | 0.49±0.10 | 0.51±0.03 |

| | | | | | | | |
|---|---|---|---|---|---|---|---|
| j | 20.47, 13.10 | 12:33:00 | 32.69 | 1614±3 | 18±6 | 0.19±0.10 | 0.50±0.10 | 0.51±0.03 |
| k | 20.47, 13.11 | 12:35:30 | 33.11 | 1616±3 | 17±5 | 0.19±0.10 | 0.52±0.10 | 0.52±0.03 |
| l | 20.47, 13.11 | 12:37:58 | 33.54 | 1615±3 | 16±4 | 0.19±0.10 | 0.52±0.10 | 0.51±0.03 |
| m | 20.47, 13.11 | 12:40:28 | 33.97 | 1614±3 | 17±6 | 0.19±0.10 | 0.52±0.10 | 0.52±0.03 |
| n | 20.47, 13.11 | 12:45:25 | 34.85 | 1615±1 | 25±10 | 0.17±0.08 | 0.47±0.10 | 0.51±0.03 |
| o | 20.46, 13.12 | 12:47:55 | 35.30 | 1614±2 | 28±11 | 0.17±0.08 | 0.45±0.11 | 0.50±0.03 |
| p | 20.46, 13.13 | 12:50:23 | 35.76 | 1614±2 | 29±10 | 0.17±0.08 | 0.44±0.11 | 0.50±0.02 |

Table 2 Aerosol microphysical-optical properties of carbonaceous smoke model and radiative transfer configurations assumed in the radiative transfer simulations.

| AERONET Site | $R_\mu/R_\sigma$ | | $i_{real}$ | | $i_{img}$ | | SSA | |
|---|---|---|---|---|---|---|---|---|
| **Mongu, Zambia** | Fine | Coarse | 470 nm | 860 nm | 470 nm | 860 nm | 470 nm | 860 nm |
| | 0.0898/1.4896 | 0.9444/1.9326 | 1.50 | 1.50 | 0.0262 | 0.0248 | 0.85 | 0.79 |

*Aerosol and Geometry Configuration in RT calculations*
*Aerosol optical depth nodes [500 nm]: [0.0, 0.1, 0.2, 0.3, 0.4, 0.5, 0.7]*
*Extinction Angstrom Exponent: 1.77*
*Aerosol Layer Height for above-cloud aerosols: 1.0-1.5 km uniform profile*
*Aerosol Layer Height for above-aircraft aerosols: 1.75-3.75 km uniform profile*

*Solar Zenith Angle: [0, 10, 20, 30, 40, 50, 60]*
*Viewing Zenith Angle: [0, 6, 12, 18, 24, 30, 36, 42, 48, 54, 60, 66, 72, 80]*
*Relative Azimuth Angle: [0, 20, 40, 60, 80, 100, 120, 140, 160, 180]*

Table 3. Key parameters of the simulations used for exploring the impact of three-dimensional radiative processes.

| Parameter | Value |
|---|---|
| Aircraft altitude | 1.6 km |
| Cloud base and top altitudes | 0.5 km, 1 km |
| Base and top altitudes of homogeneous aerosol layer | 1 km, 2.5 km |
| Cloud optical depth (COD) | Linear decrease from the edge to the center line of a 300 m wide and infinitely long trough. Outside trough: COD = 17; center line of trough: COD = 7 or 4.7. |
| Cloud droplet effective radius | 10 μm |
| Aerosol optical depth at 0.5 μm | Above CAR: 0.5; below CAR: 0.35 (0 in some tests) |
| Aerosol size distribution | Small mode of MODIS absorbing smoke model in Levy et al. (2007) |
| Aerosol absorption | Refractive index: $1.5 + i*0.033$. Resulting single scattering albedos: 0.85 at 0.47 μm and 0.79 at 0.87 μm |
| Surface albedo | 0.05 |
| Solar zenith angle | 33° |
| Viewing zenith angle | 0° |

Table 4: Measured spectral albedo (with atmosphere) for each BRDF case.

| Case | Wavelength (μm) | | | | | | | |
|------|-------|-------|-------|-------|-------|-------|-------|-------|
|      | 0.340 | 0.381 | 0.472 | 0.682 | 0.870 | 1.036 | 1.219 | 1.273 |
| a | 0.32 | 0.38 | 0.41 | 0.45 | 0.46 | 0.47 | 0.41 | 0.40 |
| b | 0.34 | 0.40 | 0.44 | 0.48 | 0.49 | 0.51 | 0.44 | 0.43 |
| c | 0.32 | 0.38 | 0.41 | 0.45 | 0.46 | 0.47 | 0.41 | 0.40 |
| d | 0.25 | 0.30 | 0.31 | 0.33 | 0.34 | 0.35 | 0.31 | 0.30 |
| e | 0.22 | 0.26 | 0.28 | 0.30 | 0.31 | 0.32 | 0.28 | 0.27 |
| f | 0.23 | 0.26 | 0.27 | 0.30 | 0.31 | 0.31 | 0.27 | 0.27 |
| g | 0.23 | 0.27 | 0.27 | 0.31 | 0.31 | 0.32 | 0.27 | 0.27 |
| h | 0.42 | 0.51 | 0.54 | 0.60 | 0.62 | 0.64 | 0.55 | 0.53 |
| i | 0.40 | 0.48 | 0.52 | 0.57 | 0.58 | 0.61 | 0.52 | 0.50 |
| j | 0.40 | 0.47 | 0.51 | 0.56 | 0.57 | 0.60 | 0.51 | 0.49 |
| k | 0.39 | 0.47 | 0.50 | 0.55 | 0.56 | 0.58 | 0.50 | 0.49 |
| l | 0.39 | 0.47 | 0.50 | 0.55 | 0.57 | 0.59 | 0.50 | 0.49 |
| m | 0.40 | 0.48 | 0.51 | 0.57 | 0.58 | 0.60 | 0.52 | 0.50 |
| n | 0.45 | 0.55 | 0.59 | 0.65 | 0.68 | 0.70 | 0.59 | 0.57 |

| | | | | | | | | |
|---|---|---|---|---|---|---|---|---|
| o | 0.47 | 0.57 | 0.62 | 0.69 | 0.71 | 0.73 | 0.61 | 0.59 |
| p | 0.49 | 0.59 | 0.64 | 0.71 | 0.73 | 0.75 | 0.62 | 0.61 |

Table 5: Simulated CAR BRFs at the center of a hypothetical trough.

| | $BRF_{0.47\,\mu m}$ | $BRF_{0.87\,\mu m}$ | $BRF_{0.47\,\mu m}\,/\,BRF_{0.87\,\mu m}$ |
|---|---|---|---|
| No BCA, COD=7.0, 1D | $0.28861 \pm 0.00007$ | $0.34162 \pm 0.00007$ | $0.84483 \pm 0.00038$ |
| No BCA, COD=7.0, 3D | $0.35663 \pm 0.00008$ | $0.42296 \pm 0.00008$ | $0.84318 \pm 0.00035$ |
| No BCA, COD=4.7, 3D | $0.28829 \pm 0.00008$ | $0.34243 \pm 0.00008$ | $0.84189 \pm 0.00044$ |
| Yes BCA, COD=7.0, 1D | $0.25203 \pm 0.00004$ | $0.32416 \pm 0.00006$ | $0.77749 \pm 0.00027$ |
| Yes BCA, COD=7.0, 3D | $0.31018 \pm 0.00006$ | $0.40075 \pm 0.00007$ | $0.77400 \pm 0.00028$ |
| Yes BCA, COD=4.7, 3D | $0.25037 \pm 0.00005$ | $0.32414 \pm 0.00006$ | $0.77241 \pm 0.00030$ |

**Figure 1:** Location of the measurements. On 13 September 2000, the NASA's Cloud Absorption Radiometer (CAR) on board the University of Washington Convair-580 research aircraft obtained measurements over marine stratocumulus offshore of Namibia at several locations marked by the aircraft ground track on the map inset. The aircraft completed multiple circular flight tracks (>16) at different locations, shown on the enlarged map of the rectangular box area,  and labelled alphabetically, a-p, based on the time of observations (see Table 1). The

595 circular flight tracks were performed primarily for the airborne measurements of bidirectional reflectance distribution function (BRDF) (cases a-d and h-p), and in a few instances (cases e-g) represent vertical profiles for physical and chemical measurements. The marine stratus clouds were extensive as seen by the MODIS/Terra instrument on the same day around 09:25 UTC (see the map inset). The CV-580 flight began just prior to 10:00 UTC and ended at 13:00 UTC.  The enlarged map is derived from GWELD product generated browse image (Roy and Zhang 2019).

**Figure 2: (**a). The University of Washington's Convair-580 research aircraft in Pietersburg, South Africa, for SAFARI 2000. (b) Schematic of NASA's Cloud Absorption Radiometer (CAR), which was mounted in the nose of the CV 580 aircraft. (c) A cumulonimbus cloud observed with CAR during Flight No. 2034 on September 14, 2011, 18:35 - 18:40 UTC, in Florida to illustrate the kind of images acquired by CAR. (d) Specifications for the CAR, which contains 14 narrow spectral bands between 0.34 and 2.30 μm.

**Figure 3:** CAR quicklook image (constructed from three bands at 1.04, 0.87, and 0.47 μm) obtained over the marine stratocumulus clouds.

The circular flight track by the aircraft allows the CAR to image the sky and surface in all viewing zenith and azimuthal angles, and covering an area defined by a diameter of about 4 km on the surface (assuming the aircraft is flying 600 m above the surface). The unique feature of these measurements is the solar disks, which define the start and end point for each circle. A prominent feature of the marine stratocumulus clouds is the presence of the cloud bow ring associated with scattering by water droplets and with a peak at ~75° zenith angle in the antisolar direction.

**Figure 4**: Measured angular distribution of sky radiance (a & c) and cloud reflected radiance (b & d) at selected wavelengths (λ=0.682 μm and λ=0.874 μm) obtained at about 12:47:55 UTC with a solar zenith angle of ~35.30° (Table 1: case o). The measured (sky or surface) radiance in any given direction is normalized by the solar irradiance incident on the top of the atmosphere, assuming mean Sun–Earth distance, and then converted to a non-dimensional quantity  equivalent to effective BRF (or BRDF times π). The view zenith angle (θ) on the polar plots is represented as the radial distance from the center (0°) towards the periphery (90°) and the azimuthal angle (φ) as the arc

length from the solar principal plane (0° ≤ φ ≤ 360°). The principal plane is within the 0° – 180° azimuthal plane (the vertical plane passing through the solar position). Figures 4e & 4f show measured radiance at eight CAR spectral bands (0.34-1.27 μm) (sky and clouds) at a constant view zenith angle (50°) at different azimuthal planes angled 0°, 45°, 90°, 135°, 180°, 225°, 270° and 315°.

**Figure 5:** BRF at λ=0.472 μm for different solar zenith angles (23°< SZA <34°) and cloud optical thickness. The marine stratocumulus are often extensive and flat, but contain areas that have thinner clouds or even open cells that allows radiation to penetrate through and therefore

have lower BRF values as shown by the blue colors. A prominent feature of the marine stratocumulus clouds is the presence of the cloud bow ring associated with scattering by water droplets and with a peak at ~75° zenith angle in the antisolar direction.

**Figure 6:** BRF at 0.874 μm obtained at different solar zenith angles (23°< SZA <34°) and locations over the marine stratocumulus off the Skeleton coastline in Namibia for the 16 cases described in Table 1. A prominent feature of the marine stratocumulus clouds is the presence of the cloud bow ring associated with scattering by water droplets and with a peak at ~75° zenith angle in the antisolar direction.

**Figure 7:** Spectral albedo (with atmosphere) for all the 16 cases at λ=0.470 μm and λ= 0.870 μm.

**Figure 8:** Retrieved aerosol optical depth (λ= 0.500 μm) above clouds and the aircraft, obtained from the CAR sky radiance measurements. Note that the actual retrievals are performed at 470 nm and 860 nm assuming an Extinction Angstrom Exponent of 1.77 (see also Table 2). Pixels without valid retrievals are shaded white. The spurious retrieval of AOD around the solar disk is a result of saturation in the CAR reflectance measurements and partly due to the inability of the RT model in simulating reflectance when directly looking at the Sun.

**Figure 9:** Retrieved aerosol optical depth(λ= 0.500 μm)  above clouds and below the aircraft (AOD_cloudtop). Note that the actual retrievals are performed at 470 nm and 860 nm assuming an Extinction Angstrom Exponent of 1.77 (see also Table 2). Pixels without valid retrievals are shaded white.

**Figure 10:** Retrieved cloud optical depth. Pixels without valid retrievals are shaded white.

**Figure 11**: Scatter plot ACAOD vs COD for view zenith angles 0°-30° (blue color dots), 30°-60° (green color dots), and 60°-90° (red color dots)**.**

**Figure 12:** Comparison of the retrieved parameters averaged over all the viewing directions for each case (a-p).

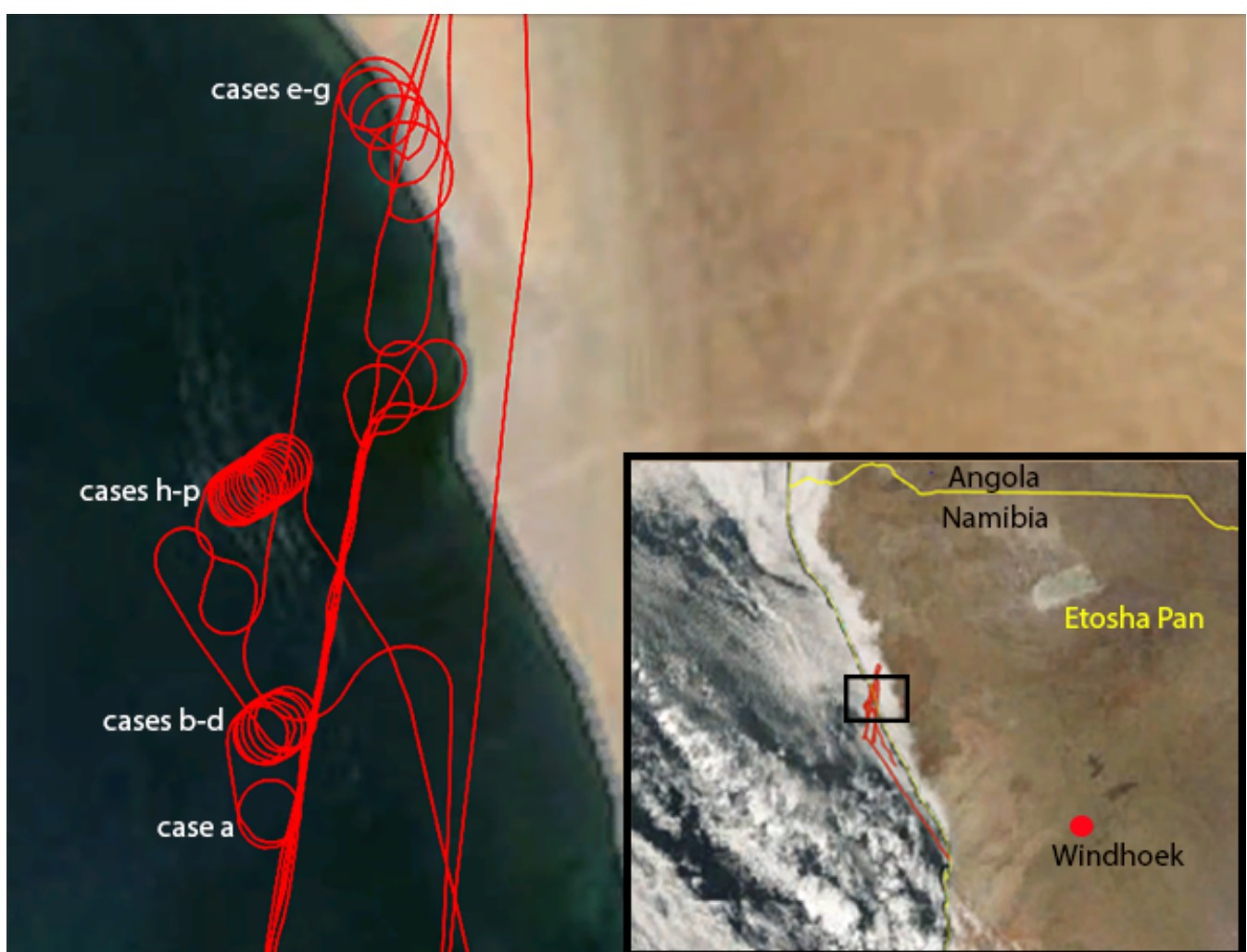

Figure 1: Location of the measurements. On 13 September 2000, the NASA's Cloud Absorption Radiometer (CAR) on board
the University of Washington Convair-580 research aircraft obtained measurements over marine stratocumulus
offshore of Namibia at several locations marked by the aircraft ground track on the map inset. The aircraft completed
multiple circular flight tracks (>16) at different locations, shown on the enlarged map of the rectangular box area (box
a), and labelled alphabetically, a-p, based on the time of observations (see Table 1). The circular flight tracks were
performed primarily for the airborne measurements of bidirectional reflectance-distribution function (BRDF) (cases
a-d and h-p), and in a few instances (cases e-g) represent vertical profiles for physical and chemical measurements.
The marine stratus clouds were extensive as seen by the MODIS/Terra instrument on the same day around 09:25 UTC
(see the map inset). The CV-580 flight began just prior to 10:00 UTC and ended at 13:00 UTC. The enlarged map is
derived from GWELD product generated browse image (Roy and Zhang 2019).

**(a)** University of Washington Convair CV-580

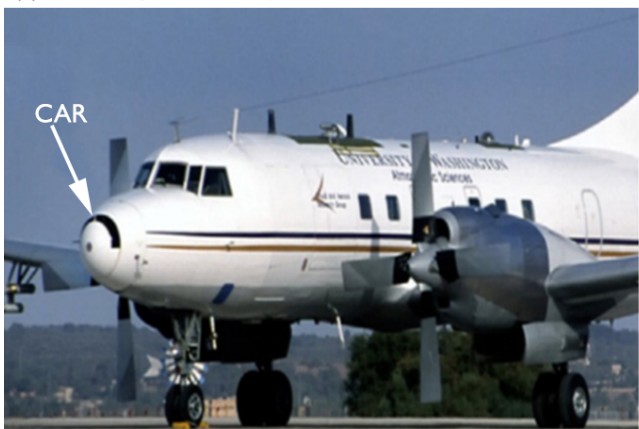

CAR

**(c)** CAR quicklook image of a Cumulonimbus Cloud

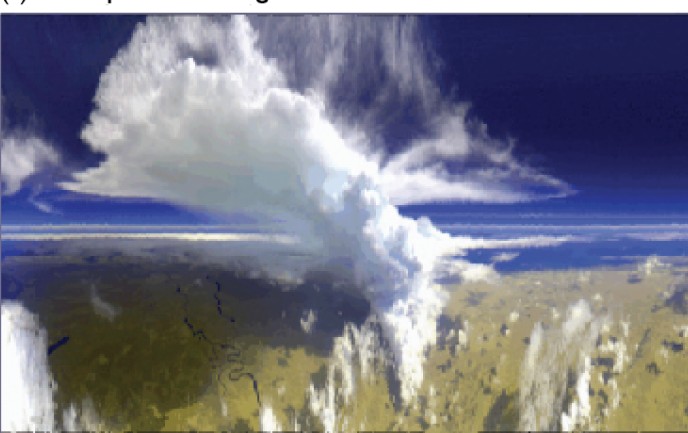

**(b)** CAR Schematic

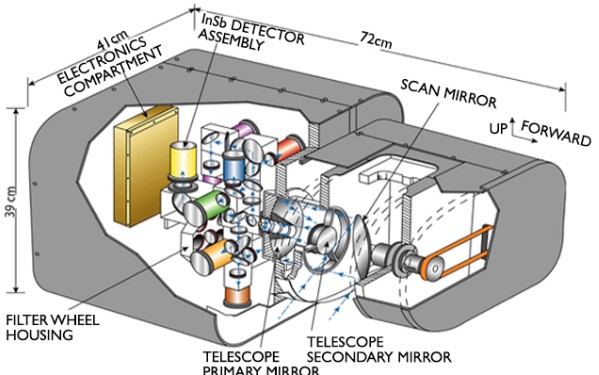

**(d)** Cloud Absorption Radiometer (CAR) Specifications

| | |
|---|---|
| Angular Scan Range | 190° |
| Instantaneous field of view | 17.5 mrad(1°) |
| Pixels per scan line | 382 |
| Scan rate | 1.67 lines per second (100 rpm) |
| Spectral channels (μm): bandwidth (FWHM) | 14 (8 continuously sampled and last six in filter wheel: 034(0.009), 0.380(0.006), 0.472(0.021) 0.683(0.021), 0.871(0.022), 1.037(0.021), 1.222(0.023), 1.275(0.023), 1.564(0.032), 1.657(0.042), 1.738(0.040), 2.105(0.045), 2.202(0.043), 2.303(0.044) |

Figure 2: (a). The University of Washington's Convair-580 research aircraft in Pietersburg, South Africa, for SAFARI 2000. (b) Schematic of NASA's Cloud Absorption Radiometer (CAR), which was mounted in the nose of the CV 580 aircraft. (c) A cumulonimbus cloud observed with CAR during Flight No. 2034 on September 14, 2011, 18:35 - 18:40 UTC, in Florida to illustrate the kind of images acquired by CAR. (d) Specifications of CAR, which contains 14 narrow spectral bands between 0.34 and 2.30 μm.

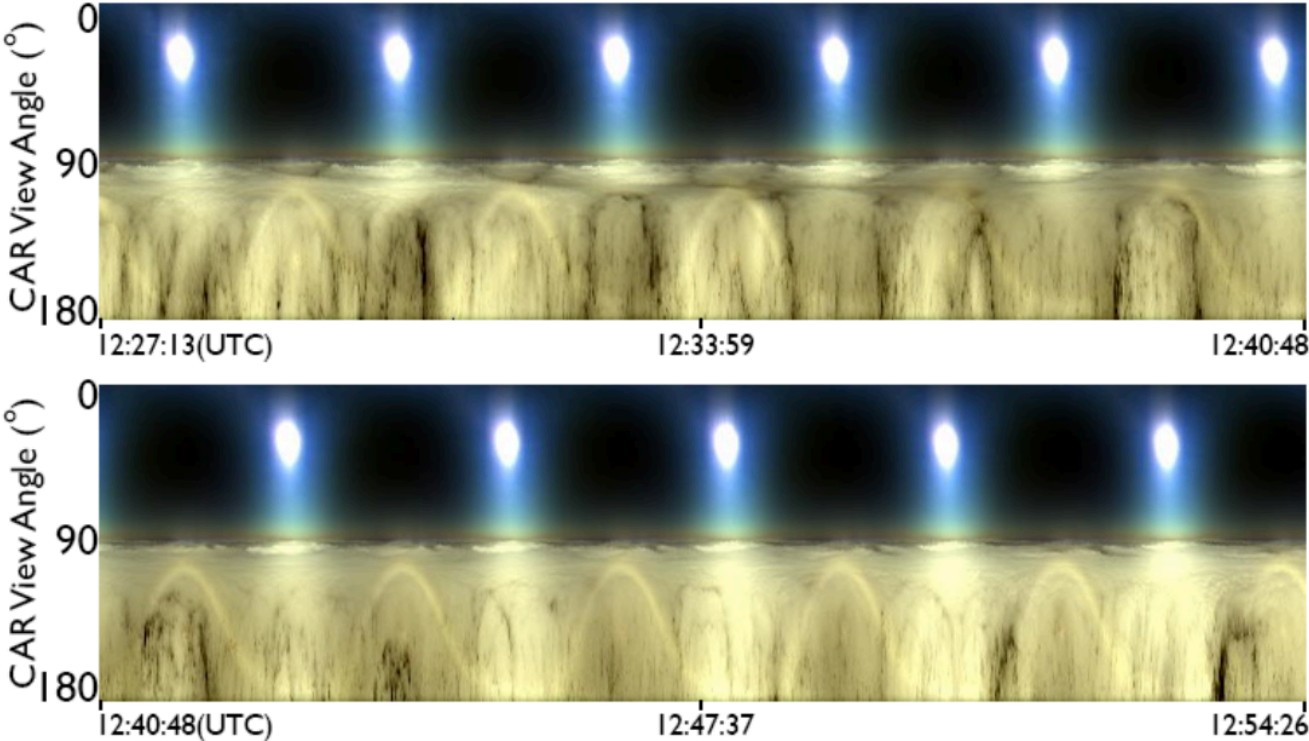

Figure 3: CAR quicklook image (constructed from three bands at 1.04, 0.87, and 0.47 µm) obtained over the marine stratocumulus clouds. The circular flight track by the aircraft allows the CAR to image the sky and surface in all viewing zenith and azimuthal angles, and covering an area defined by a diameter of about 4 km on the surface (assuming the aircraft is flying 600 m above the surface). The unique feature of these measurements is the solar disks, which define the start and end point for each circle. A prominent feature of the marine stratocumulus clouds is the

presence of the cloud bow ring associated with scattering by water droplets and with a peak at ~75° zenith angle in the antisolar direction.

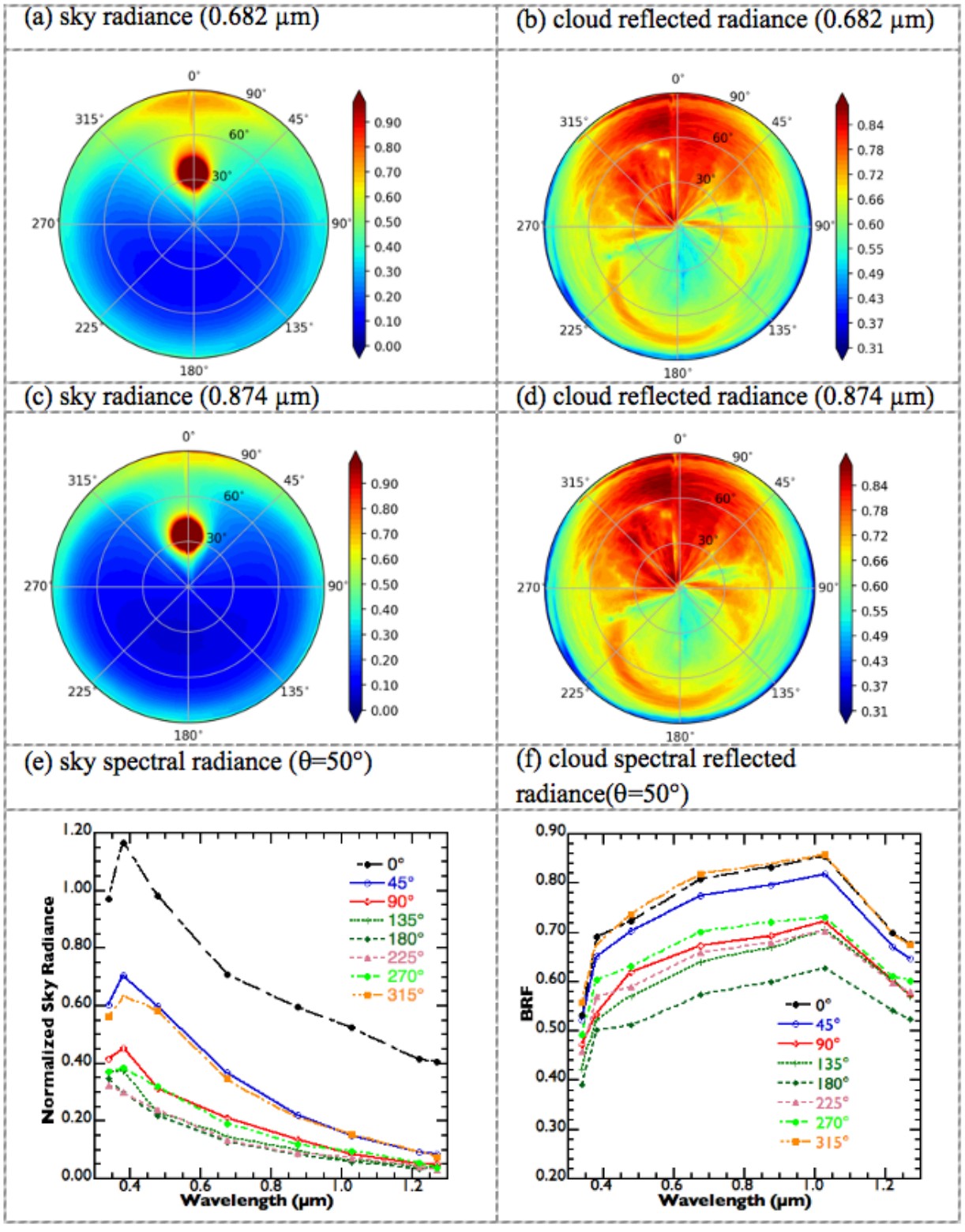

Figure 4: Measured angular distribution of sky radiance (a & c) and cloud reflected radiance (b & d) at selected wavelengths

(λ=0.682 μm and λ=0.874 μm) obtained at about 12:47:55 UTC with a solar zenith angle of ~35.30° (Table 1: case

o). The measured (sky or surface) radiance in any given direction is normalized by the solar irradiance incident on

the top of the atmosphere, assuming mean Sun–Earth distance, and then converted to a non-dimensional quantity

equivalent to effective BRF (or BRDF times $\pi$). The view zenith angle ($\theta$) on the polar plots is represented as the

radial distance from the center (0°) towards the periphery (90°) and the azimuthal angle ($\varphi$) as the arc length from the

solar principal plane ($0° \leq \varphi \leq 360°$). The principal plane is within the $0° - 180°$ azimuthal plane (the vertical plane

passing through the solar position). Figures 4e & 4f show measured radiance at eight CAR spectral bands (0.34-1.27

μm) (sky and clouds) at a constant view zenith angle (50°) at different azimuthal planes angled 0°, 45°, 90°, 135°,

180°, 225°, 270° and 315°.

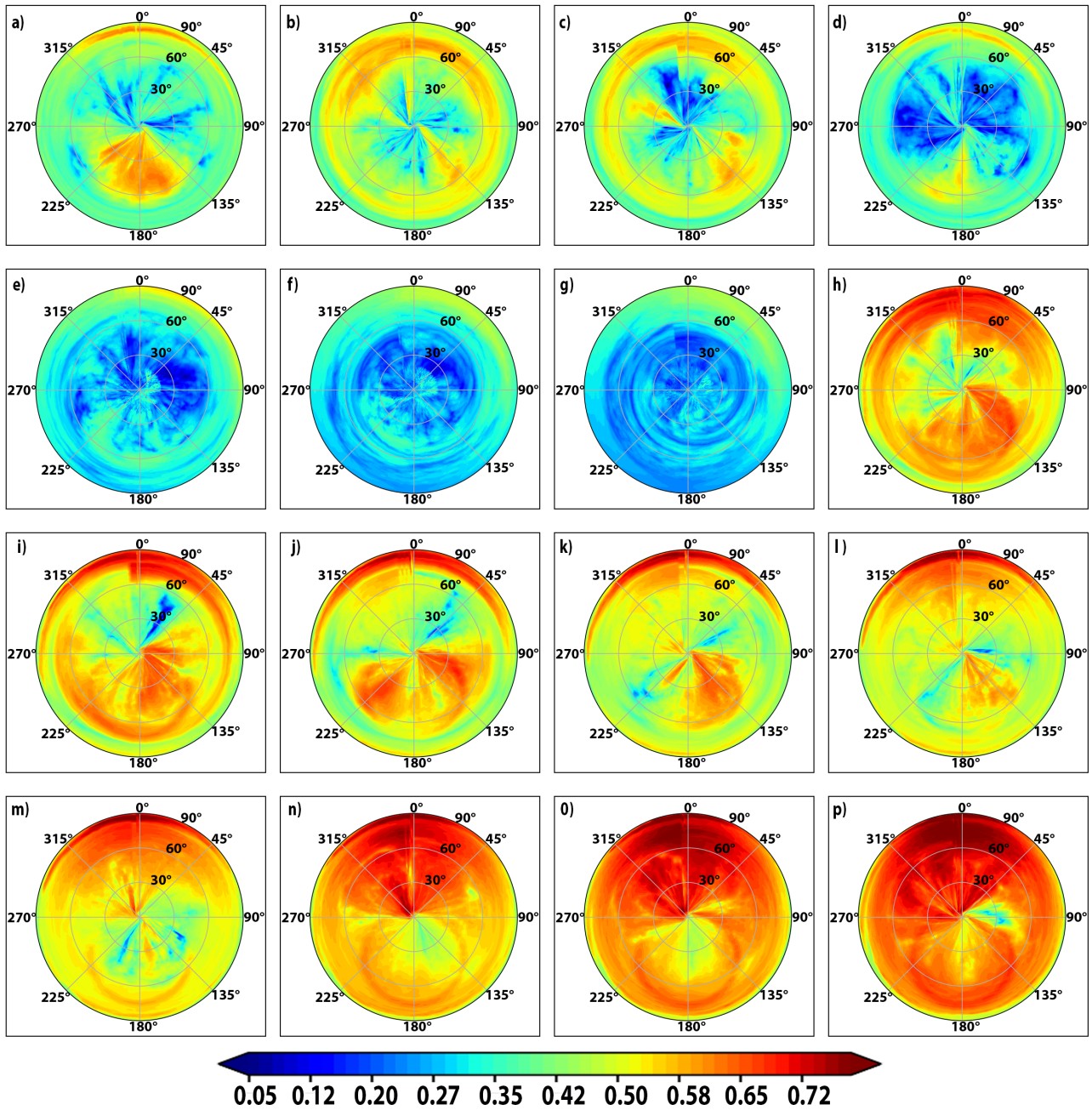

Figure 5: BRF at λ=0.472 μm for different solar zenith angles (23°< SZA <34°) and cloud optical thickness. The marine stratocumulus are often extensive and flat, but contain areas that have thinner clouds or even open cells that allows radiation to penetrate through and therefore have lower BRF values as shown by the blue colors. A prominent feature

of the marine stratocumulus clouds is the presence of the cloud bow ring associated with scattering by water droplets and with a peak at ~75° zenith angle in the antisolar direction.

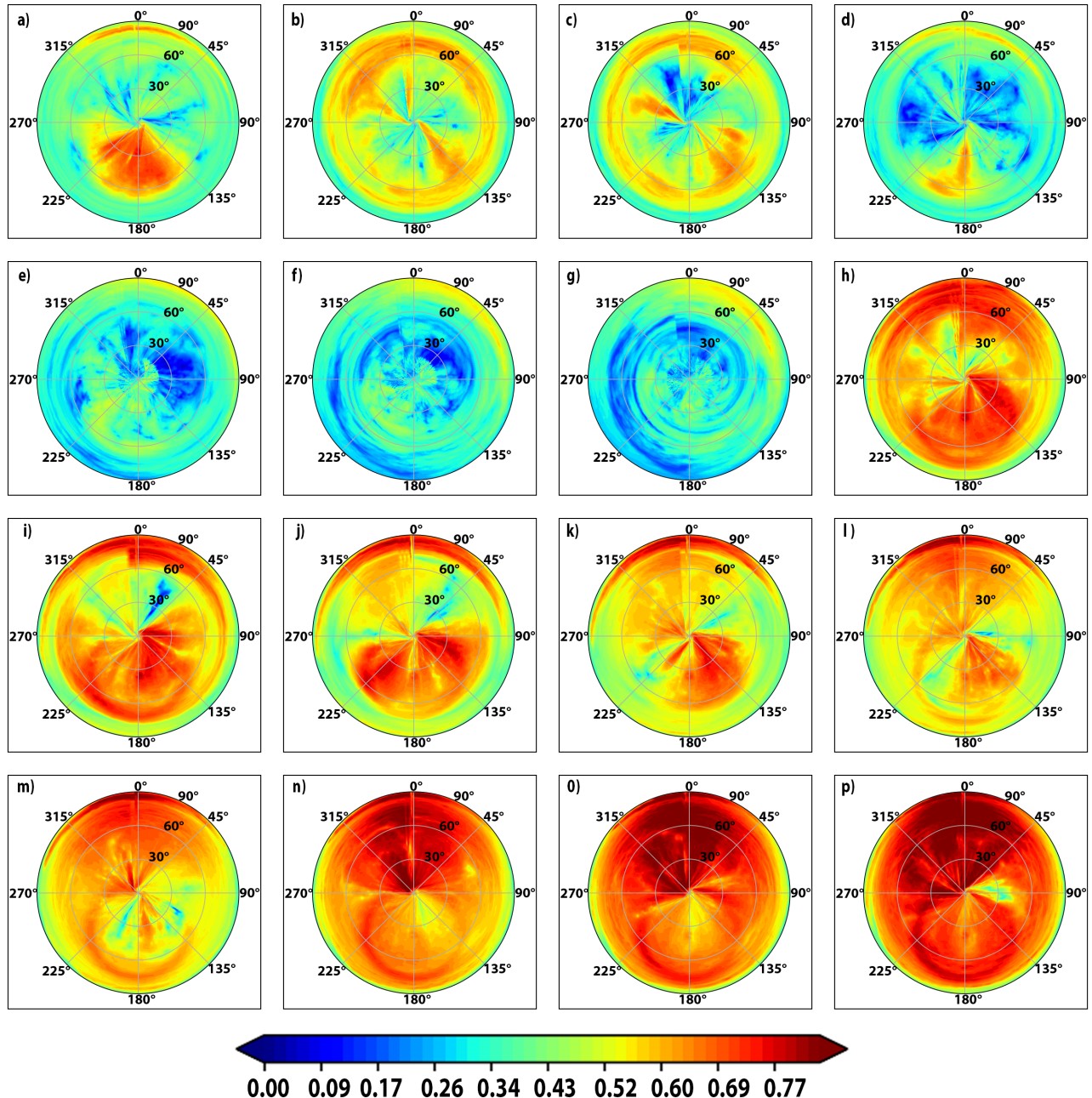

Figure 6: BRF at 0.874 μm obtained at different solar zenith angles (23°< SZA <34°) and locations over the marine stratocumulus off the Skeleton coastline in Namibia for the 16 cases described in Table 1. A prominent feature of the marine stratocumulus clouds is the presence of the cloud bow ring associated with scattering by water droplets and with a peak at ~75° zenith angle in the antisolar direction.

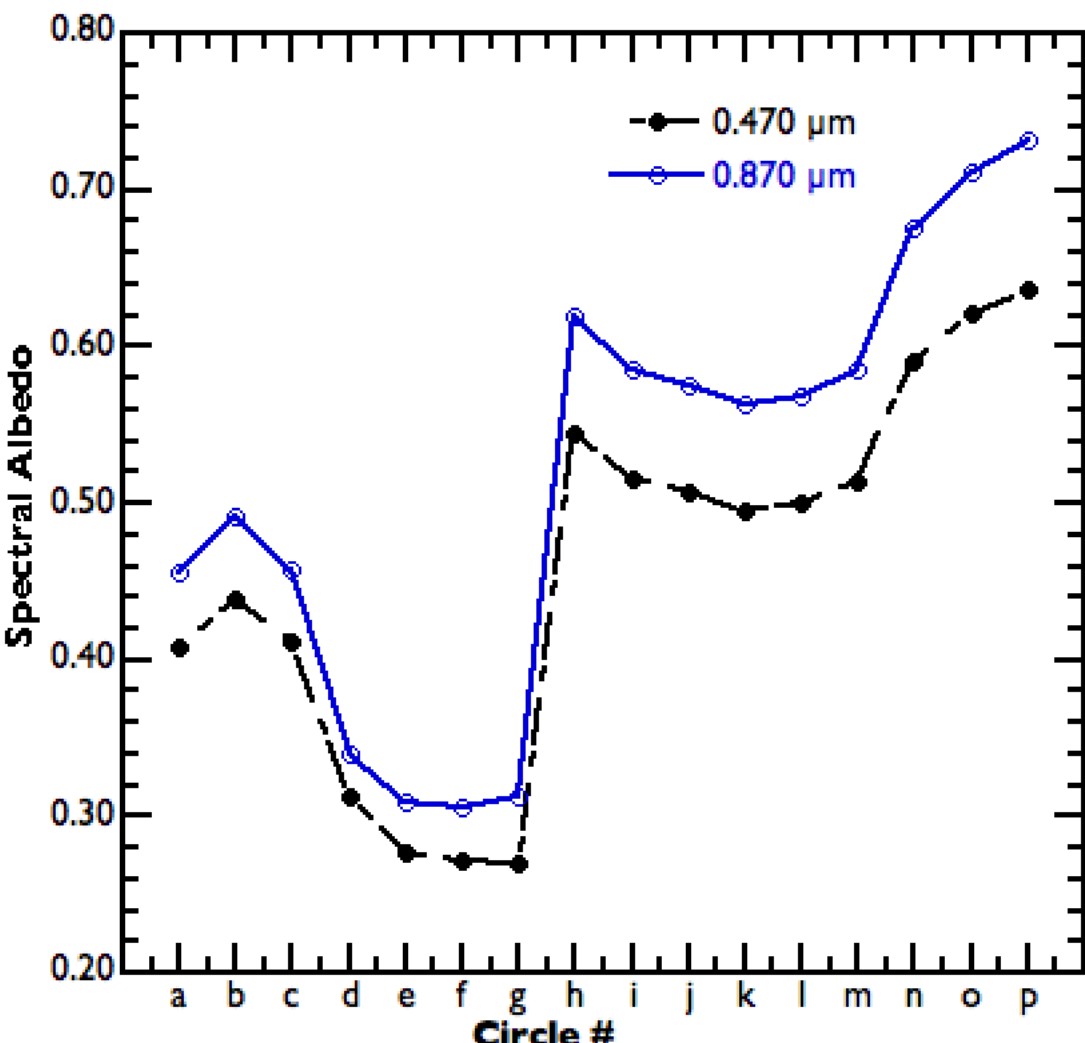

Figure 7: Spectral albedo (with atmosphere) for all the 16 cases at λ=0.470 μm and λ= 0.870 μm.

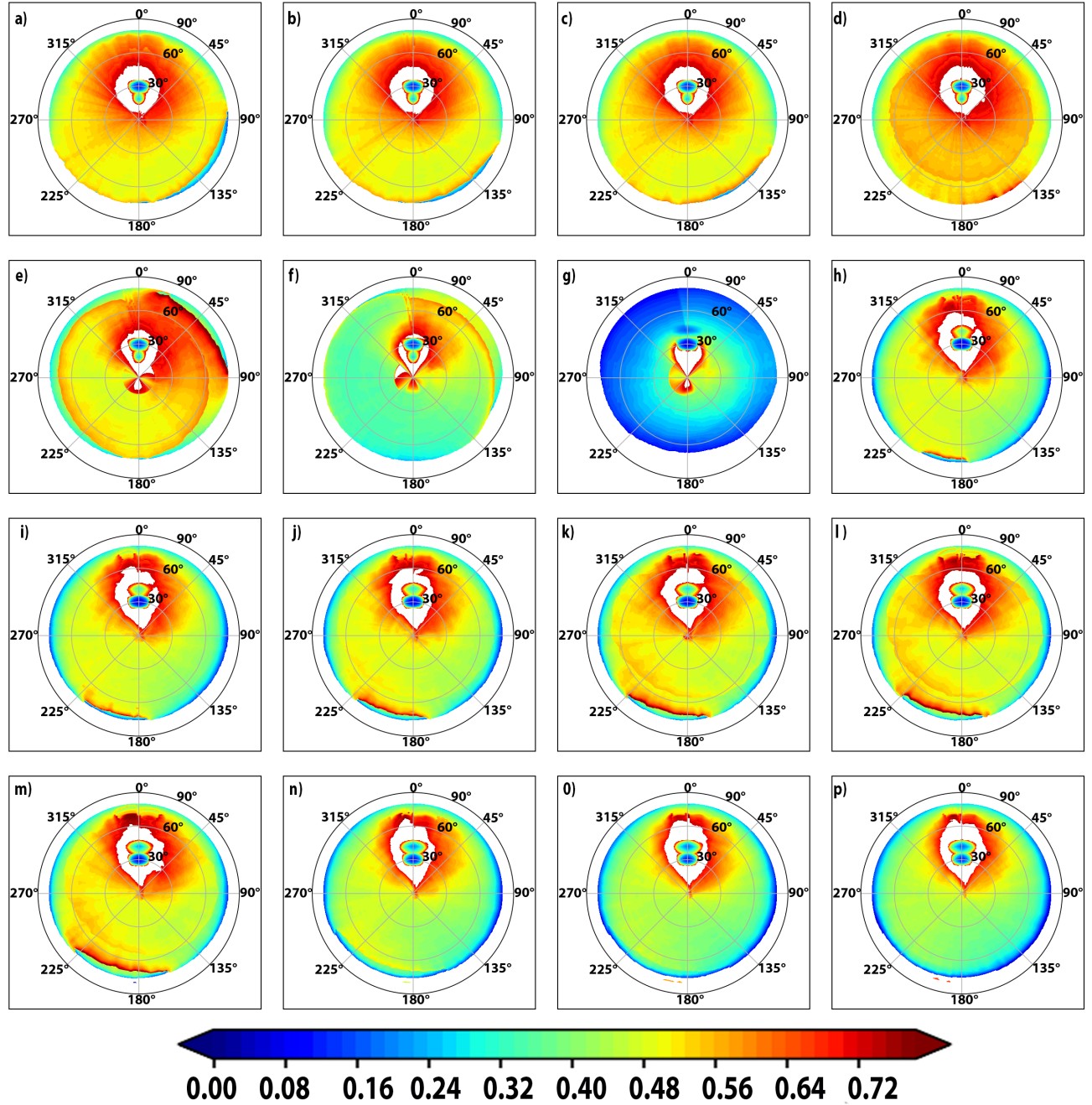

Figure 8: Retrieved aerosol optical depth (λ= 0.500 μm) above clouds and the aircraft, obtained from the CAR sky radiance measurements. Note that the actual retrievals are performed at 470 nm and 860 nm assuming an Extinction Angstrom Exponent of 1.77 (see also Table 2). Pixels without valid retrievals are shaded white. The spurious retrieval of AOD

around the solar disk is a result of saturation in the CAR reflectance measurements and partly due to the inability of the RT model in simulating reflectance when directly looking at the Sun.

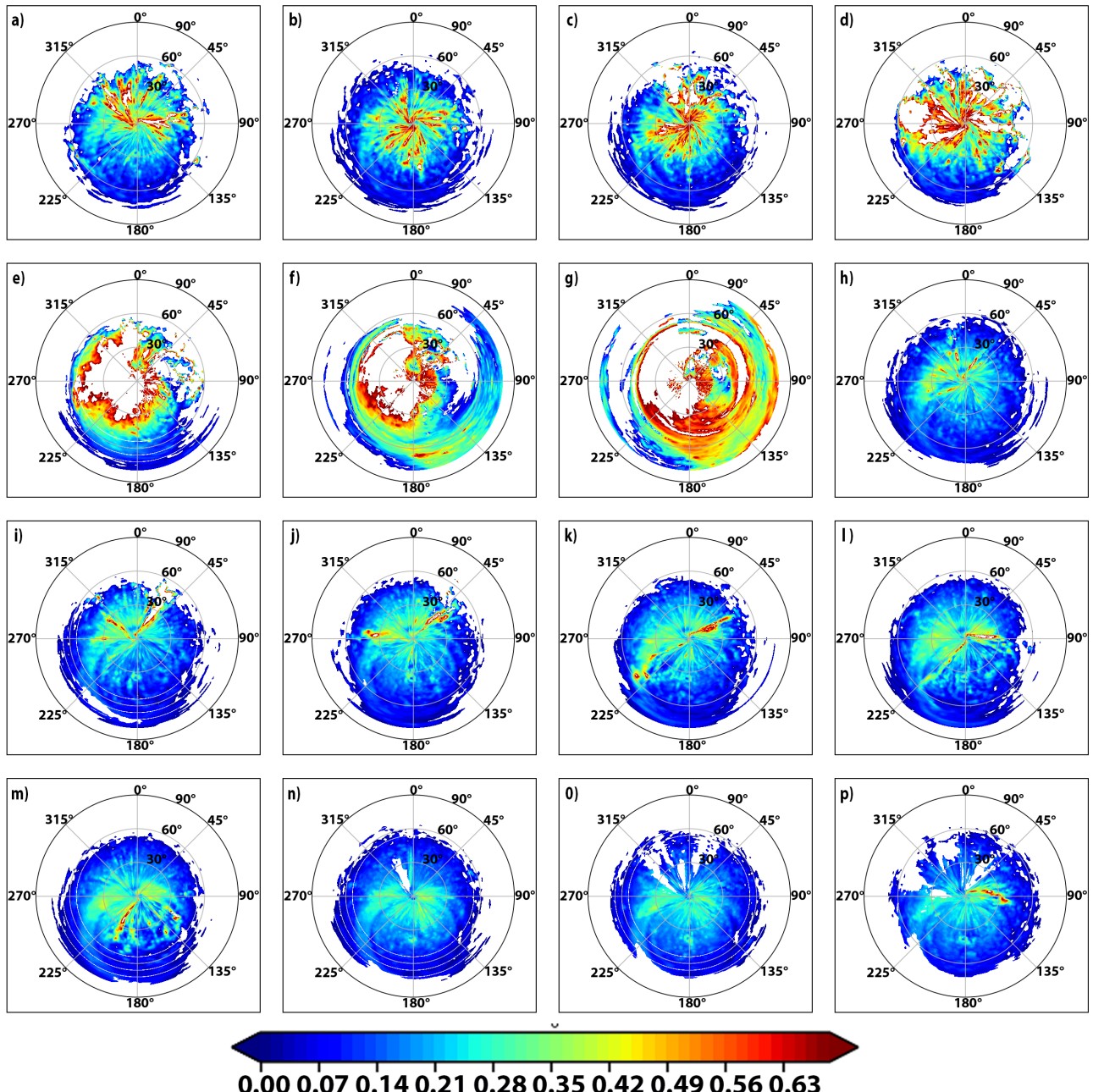

Figure 9: Retrieved aerosol optical depth($\lambda= 0.500$ μm) above clouds and below the aircraft (AOD_cloudtop). Note that the actual retrievals are performed at 470 nm and 860 nm assuming an Extinction Angstrom Exponent of 1.77 (see also Table 2). Pixels without valid retrievals are shaded white.

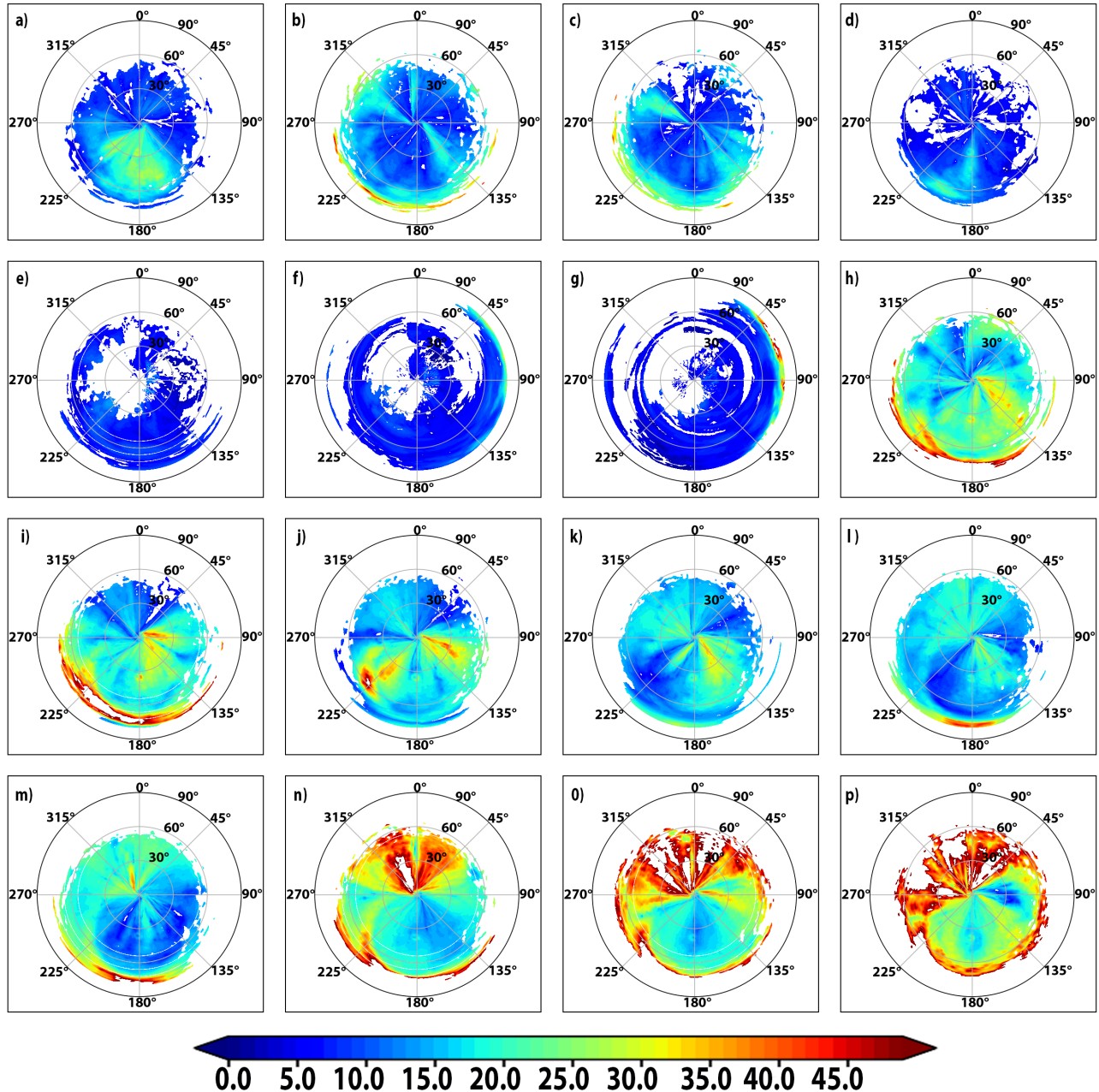

Figure 10: Retrieved cloud optical depth. Pixels without valid retrievals are shaded white.

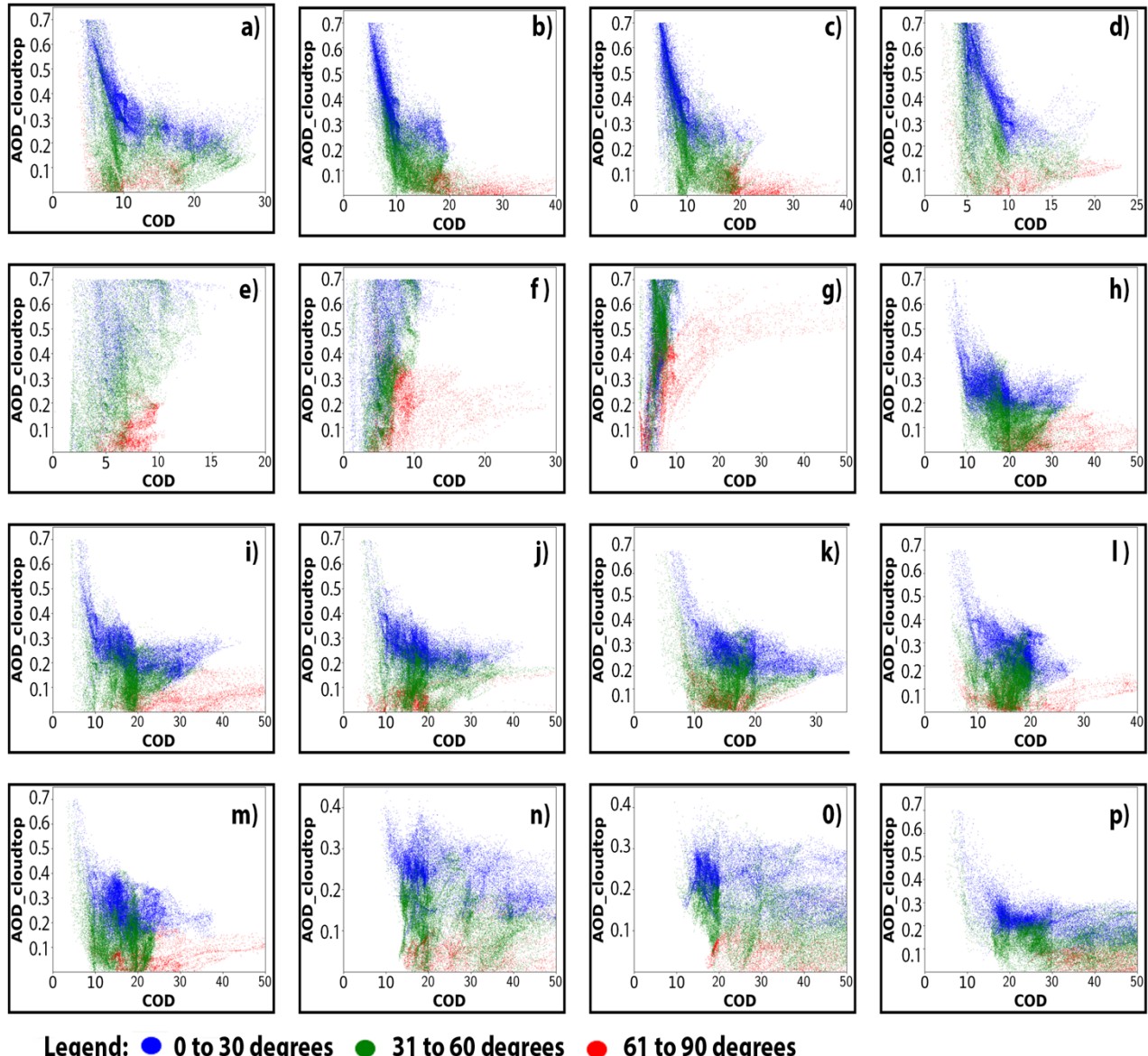

Figure 11: Scatter plot ACAOD vs COD for view zenith angles 0°-30° (blue color dots), 30°-60° (green color dots), and 60°-90° (red color dots).

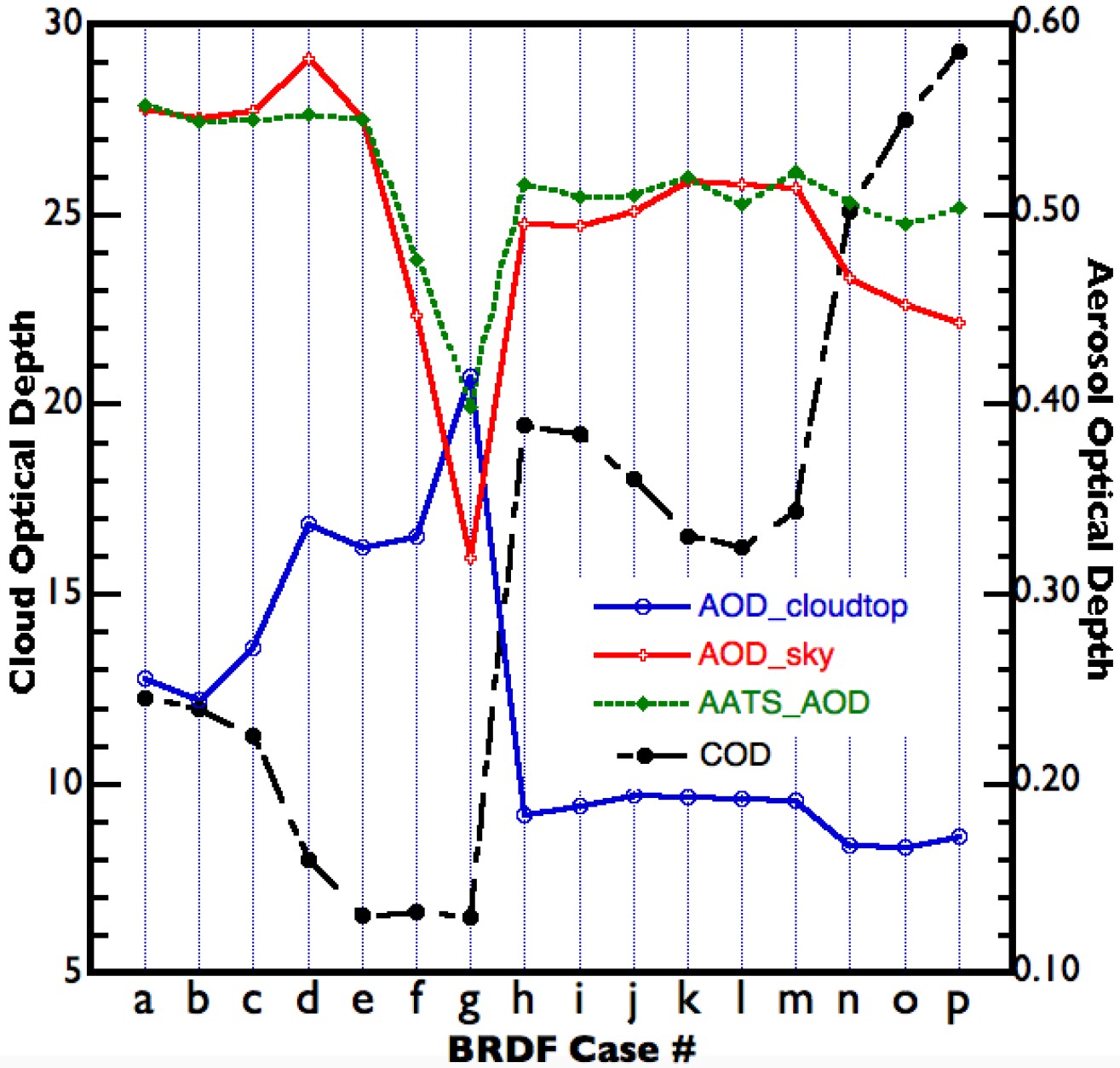

Figure 12: Comparison of the retrieved parameters averaged over all the viewing directions for each case (a-p).
