# Peer review of "A new measurement approach for validating satellite-based above cloud aerosol optical depth"

_Atmospheric Measurement Techniques, 2020_

## Referee Comment (RC1) · Anonymous Referee #1 · 21 Aug 2020

General Comments: The aim of this manuscript is to retrieve aerosol optical depth above clouds using a novel airborne measurement approach of simultaneously measuring scattered radiation above and below the aircraft, and thereby demonstrate an effective observational tool to validate satellite-based aerosol retrievals above clouds. The authors used NASA's Cloud Absorption Radiometer on board UW CV-580 during SAFARI-2000 field campaign off Namibia coast focusing on a case study on 13 September 2000. The major advantage of the instrument is its complete azimuth measurements of sky and reflected radiances. This capability allows above-cloud AOD retrieval for 1) above the aircraft and 2) below the aircraft, both of which were above maritime stratocumulus clouds. This work expands the capability of retrieving aerosols

below aircraft that cannot be performed on an airborne sunphotometer. The authors also qualitatively addressed the 3D effect of clouds on the above-cloud AOD retrieval and the COD retrieval. The scientific novelty are suitable for publication in AMT but needs a better structuring, textual clarification, and major improvements on the figure quality. Please see below for the specific comments and technical corrections.

Specific Comments:

Line 72: Need to cite Redemann et al. (2020)

Redemann, J., Wood, R., Zuidema, P., Doherty, S. J., Luna, B., LeBlanc, S. E., et al. (2020). An overview of the ORACLES (ObseRvations of Aerosols above CLouds and their intEractionS) project: aerosol-cloud-radiation interactions in the Southeast Atlantic basin (preprint). Atmospheric Chemistry and Physics Discussions. https://doi.org/10.5194/acp-2020-449

Line 141 – 152. "Reflectances" is mentioned frequently in this paragraph, but Figure 4 only refers to sky radiance and reflected radiance. Also, please state the radiance and BRF unit.

Line 143 – 145. "reflectances….larger by factor of >3" Does this sentence refer to Figure 4e?

Line 145. "This asymmetry…..directions" Which figure panel does it refer to ?

Line 185-284. Do the retrievals assume the same aerosol intensive properties in Table 3? It appears that Table 3 only applies to 3D effect analysis.

Line 205-207. The angle information should move to the methods section.

Line 209. "correlation" is a wrong word choice unless you provide a correlation coefficient for these comparisons. Otherwise, I would mention "A careful qualitative inspection"

Line 245- 247. Since ACAOD and COD retrieval uncertainties vary at various viewing

zenith and azimuth (i.e., scattering angle), it is not enough to rely on the uncertainty analysis of a previous study. I expect some discussions on how ACAOD uncertainties vary at different sensor angles for different assumed aerosol model, particularly on the SSA.

Line 248-250. It's unclear how the AOD value for each case are obtained when the AOD values differ at various angles as shown in Figures 8 and 9. This question ties to whether bad retrievals in heterogeneous conditions are included to compute the AOD.

Line 271-274. It's true that BRF in Fig. 6 is relatively more homogeneous for cases h-m, but retrieved CODs of these cases (Fig. 10) do not convince the homogeneity of clouds.

Line 280. "....40% higher..." If the total AOD is 0.7 and AOD-cloudtop is 0.2, then total AOD is 3.5 times higher. Please clarify my confusion.

Line 294. "retrieved COD drops by roughly 50% while the retrieved ACAOD increases by roughly 50%." This sentence needs a reference to the figure numbers.

Lines 299 – 302. This paragraph should be in the methods section. Please include citations of this simulator

Line 319. The equation does not have 1D CR values for COD=4.7, so it's unclear how this equation is solved.

Lines 320-321. Percentage bias in color-ratios do not result in the same percentage bias in ACAOD.

Line 327. "similar" is unclear to the reader. Do you mean similar differences between 0.47 and 0.87 micron?

Line 351-352. The notion about anticorrelation between AOD_cloudtop and COD for COD >10 and COD<10 is not mentioned before the conclusion. This finding needs to be addressed before the conclusion. Also, correlation coefficients need to be provided

when describing anticorrelations.

Line 353. "3D effects increase retrieved ACAOD by about 3-10%" The comment for line 320-321 applies here and applies to the abstract too.

Table 3. should be moved to the methods sections.

Figure 4. The dashed lines should only be the borders, but there are several extra dashed lines within the figure that need to be removed.

Figures 5, 6, 8, 9. 10, 11. The font sizes are too small for publication quality. I suggest the authors increase font to appropriate sizes and print out a figure to make sure they can see it properly on a hard copy paper. Each figure (except for figure 10) should only have one colorbar to avoid redundancy. Panel (i) in each figure has a red underline that should be removed.

The borders of the panel letter are not in consistent places and have inconsistent sizes. The authors need to either code the letter location or remove the border around the panel letter and make sure that the letters are in a similar relative location in each polar plots.

Figure 8, 9. At which wavelengths are ACAOD reported? How were the wavelength ACAOD conversions done, if any?

Technical Corrections:

Line 42. Remove "The"

Line 46-47. Provide acronyms for aerosol optical depth and cloud optical depth.

Line 57. "wavelength" -> wavelengths

Line 76. Remove comma

Line 78. Remove extra parenthesis

Line 89. Is it 2.303 micron as mentioned in Figure 2d?

Line 121-122. What do AATS and 4STAR stand for?

Line 127. Is it 9 or 8 channels?

Line 161. Case "P" has solar zenith angle of 35.76 degrees, so 34 degrees seems incorrect.

Line 171. Remove "the"

Line 182-183. Attach this paragraph to the previous paragraph

Line 288. Remove the website link

Figure 2c. In the figure title, change "Cumulus Nimbus" to "Cumulonimbus"

Figure 2d. bandwith => bandwidth

Figure 4. Please include units for radiance

Line 494. Spell out "BDRF"

Line 604. "Fig." -> "Figure"

Line 624. "Figure 12 " is missing

Table 1: longitude of case F is partially missing

---

## Referee Comment (RC2) · Anonymous Referee #2 · 1 Sep 2020

Summary:

This paper presents a method to quantify the aerosol optical depth above and below clouds from measurements of side scattered light by CAR instrument during SAFARI-2000, onboard the UW CV-580 research aircraft. This novel use of sky radiance and cloud reflectances in combination with the color ratio technique for retrieving aerosol optical properties above cloud seems very interesting. The manuscript presents extensively multiple measurement cases, and their related retrieval. The great agreement with an airborne sunphotometer gives much confidence to the methods and results presented.

[Figure]

The manuscript is well written and reads well. It is suggested that this manuscript is to be published with major revision. The following shows major comment, and following with minor general and specific comments.

Major Comment:

There is little mention of the scattering phase function, asymmetry parameter or otherwise which are used in the retrieval methodology of this study. The retrieved above cloud aerosol optical depth presented seem to show distinct dependence on scattering angle, which is likely a large retrieval artifact, which is not at all discussed in this manuscript. There seems to be significant non-uniform aerosol optical depth within the hemisphere that seems to be related to scattering angle (at various view zenith angles) and not to the actual aerosol plume shape (Fig. 8). Is this a remnant of an inconsistent assumption in aerosol scattering phase function, or maybe incongruent asymmetry parameter? This calls into question much of the retrieval methodology. Similar considerations are raised with the seemingly always centered high in AOD_cloudtop. Albeit the very good match with AATS, one would suspect that the asymmetry parameter, or the underlying scattering phase function may be erroneous, but on average a good approximation, with its high biases compensating for its low bias.

This variation, that could be caused by a bad scattering phase function, may also be a causal link to one of the major findings of the paper, where the cloud optical depth is anti-correlated to the above cloud aerosol optical depth.

General Comments:

1. In the introduction there should be mention, and comparison of a color ratio method for above cloud AOD by Meyer et al., 2015, that is applied to MODIS, and/or similarly from Peers et al., 2015. Additionally, there is little mention of the recent work based on the ORACLES measurements that follows from SAFARI. Potential to reference Redemann et al., 2020, and potentially LeBlanc et al., 2020.

2. Discussion of the impacts of the absorption properties of aerosol seems missing, particularly when referencing the color ratio technique in Section 2.2. Maybe a reference to the absorption properties from other radiative measurements during SAFARI; Bergstrom et al., 2003, or alternatively on the variations of the absorption as showcased by Pistone et al., 2019.

3. Presentation of the figure 12, combining the AOD_cloudtop and AOD_sky might be better suited if there is inclusion of the measurement altitude, which might help indicate the partitioning. P.9 line 261: AOD from AATS would be representative either if directly above clouds, or below all significant layer of aerosol in the event of a clear-air-slot between cloud top and the bottom of the aerosol layer. It is suggested to add this caveat. The conclusion mentions this note again, but some care can be taken by careful data selection of sunphotometer data as presented by LeBlanc et al., 2020.

Specific Comments:

4. P.4 lines 121-122, AATS and 4STAR acronyms are not defined, please define and add pertinent citations.

5. P.8 line 236: typo: 'betweent' should be 'between'

6. Table 1 shows an error value of 0.00 for much of the AATS AOD, this seems improbable and likely missing a significant digit. Additionally, there is no mention of what wavelength these AODs are reported (as compared to the retrieved ACAOD).

7. Figure 4, There are no units on the colorbars, or the title is misleading – shouldn't it be radiance values in W/m^2/nm/sr, or is it normalized radiances? If normalized radiance, it is normalized to what? The solar disc is apparently saturated, therefore if you normalize to that value, wouldn't that be misleading?

8. Figure 4 a) & c), the solar disc seems to be not centered on the scattered light plot. The 0° line does not seem to be in line with the principal plane.

9. Figure 8, the AOD above clouds retrieval at the solar disc seems drastically different

than the surrounding region outside of the non-valid region.

10. Figure 12 – the figure caption lacks the identifier 'Figure 12:'

References:

Bergstrom, R., Pilewskie, P., Schmid, B. and Russell, P. B.: Estimates of the spectral aerosol single scattering albedo and aerosol radiative effects during SAFARI 2000, J. Geophys. Res., 108(D13), 1–11, doi:10.1029/2002JD002435, 2003.

LeBlanc, S. E., Redemann, J., Flynn, C., Pistone, K., Kacenelenbogen, M., Segal-rosenheimer, M., Shinozuka, Y., Dunagan, S., Dahlgren, R. P., Meyer, K., Podolske, J., Howell, S. G., Freitag, S., Small-griswold, J., Holben, B., Diamond, M., Wood, R., Formenti, P., Piketh, S., Maggs-Kölling, G., Gerber, M. and Namwoonde, A.: Above-cloud aerosol optical depth from airborne observations in the southeast Atlantic, Atmos. Chem. Phys., 20, 1565–1590, doi:10.5194/acp-20-1565-2020, 2020.

Meyer, K., Platnick, S. and Zhang, Z.: Simultaneously inferring above-cloud absorbing aerosol optical thickness and underlying liquid phase cloud optical and microphysical properties using MODIS, J. Geophys. Res., 120(11), 5524–5547, doi:10.1002/2015JD023128, 2015.

Peers, F., Waquet, F., Cornet, C., Dubuisson, P., Ducos, F., Goloub, P., Szczap, F., Tanré, D. and Thieuleux, F.: Absorption of aerosols above clouds from POLDER/PARASOL measurements and estimation of their direct radiative effect, Atmos. Chem. Phys., 15(8), 4179–4196, doi:10.5194/acp-15-4179-2015, 2015.

Pistone, K., Redemann, J., Doherty, S., Zuidema, P., Burton, S., Cairns, B., Cochrane, S., Ferrare, R., Flynn, C., Freitag, S., Howell, S. G., Kacenelenbogen, M., LeBlanc, S., Liu, X., Schmidt, K. S., III, A. J. S., Segal-Rozenhaimer, M., Shinozuka, Y., Stamnes, S., van Diedenhoven, B., Van Harten, G. and Xu, F.: Intercomparison of biomass burning aerosol optical properties from in situ and remote-sensing instruments in ORACLES-2016, Atmos. Chem. Phys., 19, 9181–9208, doi:10.5194/acp-19-9181-2019, 2019.

---

## Author Comment (AC1) · 15 Oct 2020

Specific Comments: 1. Line 72: Need to cite Redemann et al. (2020.) Redemann, J., Wood, R., Zuidema, P., Doherty, S. J., Luna, B., LeBlanc, S. E., et al. (2020). An overview of the ORACLES (ObseRvations of Aerosols above CLouds and their intEractionS) project: aerosol-cloud-radiation interactions in the Southeast Atlantic basin (preprint). Atmospheric Chemistry and Physics Discussions. https://doi.org/10.5194/acp-2020-449

Response: The new reference was added.

[Figure]

2. Line 141 – 152. "Reflectances" is mentioned frequently in this paragraph, but Figure 4 only refers to sky radiance and reflected radiance. Also, please state the radiance and BRF unit.

Response: The "reflectance" usage in the paragraph is correct. However we have added the following text to the figure caption: The measured (sky or surface) radiance in any given direction is normalized by the solar irradiance incident on the top of the atmosphere, assuming mean Sun–Earth distance, and then converted to a non-dimensional quantity equivalent to effective BRF (or BRDF times $\pi$).

3. Line 143 – 145. "reflectances. . ..larger by factor of >3" Does this sentence refer to Figure 4e?

Response: Yes, this refers to Fig. 4e. We revised the text to read "reflectances ... larger by a factor of >2" (line 157)

4. Line 145. "This asymmetry. . ...directions" Which figure panel does it refer to ?

Response: The asymmetry refers to Fig. 4e as shown in the revised sentence (line 158).

5. Line 185-284. Do the retrievals assume the same aerosol intensive properties in Table3? It appears that Table 3 only applies to 3D effect analysis.

Response [reference section 3.2 & 3.3]: The aerosol model used for the above-cloud and sky retrieval of AOD is different from the model used for investigating 3D effects. Table 2, included at the end of this response, lists the aerosol microphysical-optical properties, along with radiative transfer configurations, assumed in the above-cloud/below-aircraft and above-aircraft aerosol retrievals.

The purpose of simulating the effects of 3D effects was to gauge an overall estimates of errors in the aerosols and cloud retrievals, instead of applying the actual inversions which would be too complicated.

6. Line 205-207. The angle information should move to the methods section.

Response: Table 2, included at the end of this response, lists the aerosol microphysical-optical properties, along with radiative transfer configurations, assumed in the above-cloud/below-aircraft and above-aircraft aerosol retrievals.

7. Line 209. "correlation" is a wrong word choice unless you provide a correlation coeffi- cient for these comparisons. Otherwise, I would mention "A careful qualitative inspec- tion"

Response: adopted the reviewer's suggestion.

8. Line 245- 247. Since ACAOD and COD retrieval uncertainties vary at various view-ing zenith and azimuth (i.e., scattering angle), it is not enough to rely on the uncertainty analysis of a previous study. I expect some discussions on how ACAOD uncertainties vary at different sensor angles for different assumed aerosol model, particularly on the SSA.

Response: Good point. We added some discussions, "Additionally, studies 9Torres et al., 2012; Jethva et al., 2018) estimated uncertainty limits in ACAOD for typical range of satellite-viewing geometry (i.e., solar zenith angle 20-40°, viewing zenith angle 0-40°, and relative azimuth angle 100-150°), while varying the single-scattering albedo and aerosol layer height. The error estimates of ACAOD, not reported in these papers though, were found to be near-stable as a function of geometry in the stated ranges. A near-uniform retrieval of sky-looking AOD (above-aircraft and clouds) shown for dif-ferent CAR profiles in Figure 8 further demonstrates the stability of the algorithm for viewing zenith range 0-60°. At slant angles >60° and around the edge of the scan, the limitation of radiative transfer calculations due to its pseudo-spherical treatment in the RT code restricts the accuracy of AOD inversion."

9. Line 248-250. It's unclear how the AOD value for each case are obtained when the AOD values differ at various angles as shown in Figures 8 and 9. This question ties to

whether bad retrievals in heterogeneous conditions are included to compute the AOD.

Response: See response to #8. Also, we added the following sentence [line 284], However, we note that no explicit cloud-screening was performed on the measurements. All measurements go through the ACA algorithm where if they fit into the retrieval domain, i.e., color ratio vs. reflectance 860 nm, then a corresponding retrieval of ACAOD and aerosol-corrected COD are obtained. It is possible that heterogeneity in aerosol and cloud fields in the observed scene can introduce uncertainty in the retrievals. For instance, a mixture of cloudy and cloud-free scenes observed in a particular measurements can affect both AOD and COD inversions."

10. Line 271-274. It's true that BRF in Fig. 6 is relatively more homogeneous for cases h-m, but retrieved CODs of these cases (Fig. 10) do not convince the homogeneity of clouds.

Response: The sentence was clarified. The revised sentence now reads: The COD associated with the marine stratocumulus clouds (cases h-m) vary between 15 and 20 (Fig. 12).

11. Line 280. ". . ..40% higher. . ." If the total AOD is 0.7 and AOD-cloudtop is 0.2, then total AOD is 3.5 times higher. Please clarify my confusion.

Response: several sentences were revised to remove the confusion. The revised sentences read: For instance, in cases h, the AOD_cloudtop is 0.18 and the Sky_AOD is 0.50, implying the total above-cloud column AOD is 0.68 or 31% higher relative to the AATS_AOD retrieval. Overall, we find AOD_cloudtop ranging between 0.18 and 0.41 from the 16 cases shown in Fig. 12, indicating a notable enhancement of the overall presence of aerosols above clouds. These observations show that a significant aerosol layer is not captured by the aircraft sunphotometer, indicating the strength and effectiveness of near-simultaneous multiangular measurements scanning the sky and surface, as demonstrated in this study using CAR measurements

12. Line 294. "retrieved COD drops by roughly 50% while the retrieved ACAOD increases by roughly 50%." This sentence needs a reference to the figure numbers.

Response: We have now included the figure reference (Fig. 10k for the COD and Fig. 9k for AOD_cloudtop).

13. Lines 299 – 302. This paragraph should be in the methods section. Please include citations of this simulator

Response: Following the suggestion, we moved this paragraph into a newly added subsection of the section on methods (Section 2.3, "Three-dimensional radiation simulations"). We also modified the ending of the paragraph, so it now includes two citations about the simulation model and remains consistent with Table 3 (now called Table 2) also being moved into the methods section (following comment #19). The text now says:

This model was validated through I3RC intercomparison experiments (e.g., Cahalan et al., 2005) and was used in several other studies (e.g., Várnai et al., 2013). The key simulation parameters are listed in Table 2; additional details and the results of the simulations are discussed in Section 3.4.

Cahalan, R. F., Oreopoulos, L., Marshak, A., Evans, K. F., Davis, A. B., Pincus, R., Yetzer, K., Mayer, B., Davies, R., Ackerman, T., Barker, H., Clothiaux, E., Ellingson, R., Garay, M., Kassianov, E., Kinne, S., Macke, A., O'Hirok, W., Partain, P., Prigarin, S., Rublev, A., Stephens, G., Szczap, F., Takara, E., Várnai, T., Wen, G., and Zhuravleva, T.: The International Intercomparison of 3D Radiation Codes (I3RC): Bringing together the most advanced radiative transfer tools for cloudy atmospheres, B. Am. Meteorol. Soc., 86, 1275–1293, 2005. Várnai, T., Marshak, A., and Yang, W.: Multi-satellite aerosol observations in the vicinity of clouds, Atmos. Chem. Phys., 13, 3899–3908, https://doi.org/10.5194/acp-13-3899-2013, 2013.

In order to make the remaining text of Section 3.4 flow more smoothly after this move,

we also adjusted slightly the beginning of the text that remained in Section 3.4, so it now says:

As discussed in Section 2.3, we examined the impact of 3D radiative effects through Monte Carlo simulations whose results are listed in Table 4. In each row of this table, . . .

14. Line 319. The equation does not have 1D CR values for COD=4.7, so it's unclear how this equation is solved.

Response: Thank you for raising this point; we see now that this part of the manuscript was not clear. Therefore, we replaced lines 316-319 by the text below (which hopefully clarifies that we don't use 1D CR values for COD=4.7):

Regarding aerosol retrievals, we first examine how 3D radiative processes affect the key signal of our ACAOD retrievals, which is the impact of below-CAR aerosols (BCAs) on the BRF(0.47 $\mu$m) / BRF(0.87 $\mu$m) color ratio (CR) values. Specifically, we compare the CR values for the BCA and noBCA cases, and check whether the CR-difference is similar in 1D and 3D radiative simulations:

((CR3D(BCA) - CR1D(no BCA)) / (CR1D(BCA) - CR1D(no BCA)) = 1.052 $\pm$ 0.02 While the calculations above used the retrieved value of COD = 7 at the center of the linear trough, we also tested whether the results change if the 3D simulations use COD=4.7 instead: ((CR3D, COD=4.7(BCA) - CR1D(no BCA)) / (CR1D(BCA) - CR1D(no BCA)) = 1.075 $\pm$ 0.02. These results indicate that 3D processes strengthen the impact of BCAs on CR values by about 3-10%.

15. Lines 320-321. Percentage bias in color-ratios do not result in the same percentage bias in ACAOD.

Response: Good point. To explore this issue, we performed additional Monte Carlo simulations and replaced the sentence, "Since CR is the key signal in our ACAOD retrievals, this implies that 3D effects are likely to increase retrieved ACAOD values by

3-10%." by the text below:

To estimate the impact of these CR changes on retrieved ACAOD values, we examined the non-linearity of the CR-ACAOD relationship using additional 1D Monte Carlo simulations. These simulations used the same setup as in Table 2, except that below-aircraft ACAOD values were increased by 20%. The simulations (identified by the subscript IBCA) gave $BRF_{IBCA}(0.47\ \mu m) = 0.24523 \pm 0.00004$ and $BRF_{IBCA}(0.87\ \mu m) = 0.32069 \pm 0.00006$, yielding $CR_{IBCA} = 0.76469 \pm 0.00027$. Comparing the impact of original and increased BCA amounts on CR gives $(CR_{IBCA} - CR_{noBCA}) / (CR_{BCA} - CR_{noBCA}) = 1.1900 \pm 0.0089$. This indicates that a 20% enhancement in ACAOD causes a 19% enhancement in the CR signal, which implies that a 10% change in CR is consistent with a 10% * 20 / 19 = 10.5% change in ACAOD. Considering the uncertainties, we can say that the 3-10% impact of 3D effects on CR values corresponds to a 3-10% impact on retrieved ACAOD values.

16. Line 327. "similar" is unclear to the reader. Do you mean similar differences between 0.47 and 0.87 micron?

Response: The word "similar" in the sentence referred to (line 332): "By performing additional simulations, we found that if we decreased COD at the center of the trough from 7 to 4.7, 3D simulations would yield 0.87 $\mu m$ BRF values around 0.32—thus resulting in hypothetical retrievals yielding COD=7 (similar to the actual CAR retrievals)". No action was taken.

17. Line 351-352. The notion about anticorrelation between AOD_cloudtop and COD for COD >10 and COD<10 is not mentioned before the conclusion. This finding needs to be addressed before the conclusion. Also, correlation coefficients need to be provided when describing anticorrelations.

Response: The finding is already discussed in subsection 3.3. But in light of this comment, we have added a sentence – line 266: Also, Figure 11 results suggest a strong anticorrelation between the AOD_cloudtop and COD for cases where COD <10,

and a weaker anticorrelation for COD >10.

18. Line 353. "3D effects increase retrieved ACAOD by about 3-10%" The comment for line 320-321 applies here and applies to the abstract too.

Response: Absolutely. Based on the changes we made in response to Comment #15, we changed the 3-10% range to 3-11% range in the conclusions section and in the abstract as well.

19. Table 3. should be moved to the methods sections.

Response: Following the suggestion, we moved Table 3 into the methods section (into Section 2.3). Because of the move, the table is now called Table 2.

20. Figure 4. The dashed lines should only be the borders, but there are several extra dashed lines within the figure that need to be removed.

Response: Lines delineating each figure panel were fixed properly.

21. Figures 5, 6, 8, 9. 10, 11. The font sizes are too small for publication quality. I suggest the authors increase font to appropriate sizes and print out a figure to make sure they can see it properly on a hard copy paper. Each figure (except for figure 10) should only have one colorbar to avoid redundancy. Panel (i) in each figure has a red underline that should be removed.

Response: We have redone all the figures as recommended.

22. The borders of the panel letter are not in consistent places and have inconsistent sizes. The authors need to either code the letter location or remove the border around the panel letter and make sure that the letters are in a similar relative location in each polar plots.

Response: the lettering of the figure panels were redone.

23. Figure 8, 9. At which wavelengths are ACAOD reported? How were the wavelength

[Figure]

ACAOD conversions done, if any?

Response: The above-cloud/below-aircraft AOD and above-aircraft sky AOD are reported at 500 nm. However, the actual retrievals are performed at 470 nm and 860 nm assuming an Extinction Angstrom Exponent of 1.77. See the attached table of aerosol models and its properties assumed in the AOD inversion. Figure 8 and 9 captions were updated to include these details.

Technical Corrections: 1. Line 42. Remove "The"Line 46-47. Provide acronyms for aerosol optical depth and cloud optical depth.

Response: done!

2. Line 57. "wavelength" -> wavelengths

Response: done!

3. Line 76. Remove comma

Response: done!

4. Line 78. Remove extra parenthesis

Response: done!

5. Line 89. Is it 2.303 micron as mentioned in Figure 2d?

Response: corrected to 2.303 micron.

6. Line 121-122. What do AATS and 4STAR stand for?

Response: added: NASA Ames Airborne Tracking Sun Photometer (AATS) and Spectrometer for Sky-Scanning, Sun-Tracking Atmospheric Research (4STAR)

7. Line 127. Is it 9 or 8 channels?

Response: corrected to 8 channels.

8. Line 161. Case "P" has solar zenith angle of 35.76 degrees, so 34 degrees seems incorrect.

Response: the upper limit of the solar zenith angle was changed to 36°.

9. Line 171. Remove "the"

Response: It's not clear whether it was necessary to remove "the" in the sentence "the same as shown in Figure 1)." No action was taken.

10. Line 182-183. Attach this paragraph to the previous paragraph

Response: We combined this paragraph with the previous one, "In the following subsections, we will examine how the surface reflectance anisotropy impacts the retrievals of the optical depth (both clouds and aerosols) using the color ratio method."

11. Line 288. Remove the website link

Response: Removed the web link (http://i3rc.gsfc.nasa.gov/Publications.html). It's no longer accessible.

12. Figure 2c. In the figure title, change "Cumulus Nimbus" to "Cumulonimbus" Figure 2d. bandwith => bandwidth

Response: Figure 2c was corrected.

13. Figure 4. Please include units for radiance

Response: The measured (sky or surface) radiance in any given direction is normalized by the solar irradiance incident on the top of the atmosphere, assuming mean Sun–Earth distance, and then converted to a non-dimensional quantity equivalent to effective BRF (or BRDF times $\pi$). This explanation was added to the figure caption.

14. Line 494. Spell out "BDRF"

Response: Table 1 caption, we spelt out BRDF: Bidirectional Reflectance-distribution Function.

15. Line 604. "Fig." -> "Figure"

Response: "Fig." in Figure 7 caption was rectified.

16. Line 624. "Figure 12 " is missing

Response: "Figure 12" added to the caption.

17. Table 1: longitude of case F is partially missing

Response: In Table 1, fixed the missing longitude values of case F.

Table 2 Aerosol microphysical-optical properties of carbonaceous smoke model and radiative transfer configurations assumed in the radiative transfer simulations. AERONET Site R$\mu$/R$\sigma$ ireal iimg SSA Mongu, Zambia Fine Coarse 470 nm 860 nm 470 nm 860 nm 470 nm 860 nm 0.0898/1.4896 0.9444/1.9326 1.50 1.50 0.0262 0.0248 0.85 0.79 Aerosol and Geometry Configuration in RT calculations Aerosol optical depth nodes [500 nm]: [0.0, 0.1, 0.2, 0.3, 0.4, 0.5, 0.7] Extinction Angstrom Exponent: 1.77 Aerosol Layer Height for above-cloud aerosols: 1.0-1.5 km uniform profile Aerosol Layer Height for above-aircraft aerosols: 1.75-3.75 km uniform profile

Solar Zenith Angle: [0, 10, 20, 30, 40, 50, 60] Viewing Zenith Angle: [0, 6, 12, 18, 24, 30, 36, 42, 48, 54, 60, 66, 72, 80] Relative Azimuth Angle: [0, 20, 40, 60, 80, 100, 120, 140, 160, 180]

[Figure]

**Fig. 1.** Scattering phase function F11 of the carbonaceous aerosol model assumed in the aerosol inversion

---

## Author Comment (AC2) · 15 Oct 2020

Major Comment: 1. There is little mention of the scattering phase function, asymmetry parameter or otherwise which are used in the retrieval methodology of this study.

Response: The attached figure shows the scattering phase function F11, along with optical properties of carbonaceous aerosol model assumed in the ACAOD and sky AOD inversion.

2. The retrieved above cloud aerosol optical depth presented seem to show distinct dependence on scattering angle, which is likely a large retrieval artifact, which is not at

all discussed in this manuscript.

Response: The RT model (VLIODRT) used to create aerosol look-up table treats the outgoing radiance in a pseudo-spherical geometry. Therefore, it is expected that the aerosol radiance simulation at slant geometry, i.e., viewing zenith angle > 70° may not carry the same accuracy as the case with lower viewing angles. This may result in less accurate retrievals at extreme viewing geometries. Additionally, larger retrieval errors at lower cloud optical depth measurements and heterogeneity in aerosol and cloud fields also add to the apparent dependence on scattering angle. This discussion has now been incorporated into the manuscript (Section 2.2).

3. There seems to be significant non-uniform aerosol optical depth within the hemisphere that seems to be related to scattering angle (at various view zenith angles) and not to the actual aerosol plume shape (Fig. 8). Is this a remnant of an inconsistent assumption in aerosol scattering phase function, or maybe incongruent asymmetry parameter? This calls into question much of the retrieval methodology. Similar considerations are raised with the seemingly always centered high in AOD_cloudtop. Albeit the very good match with AATS, one would suspect that the asymmetry parameter, or the underlying scattering phase function may be erroneous, but on average a good approximation, with its high biases compensating for its low bias. This variation, that could be caused by a bad scattering phase function, may also be a causal link to one of the major findings of the paper, where the cloud optical depth is anti-correlated to the above cloud aerosol optical depth.

Response: First, the hemispheric distribution of sky AOD, i.e., retrieval above the aircraft altitude, looks more uniform throughout the scattering angle range, except around Sun disk where the CAR measurements show saturation. Second, we don't think that inconsistent assumption in aerosol scattering phase function or asymmetry parameter is a cause of remaining minor variability of AOD fields as the aerosol model used here provided a good-level of agreement between the retrieved ACAOD and AATS direct measurements [Jethva et al., 2016]. Furthermore, consistency between the sky

[Figure]

AOD retrievals from CAR measurements and that from AATS sunphotometer shown in Figure 12 of the present study (green and red dots/lines in Figure 12) stands as another supporting evidence that the retrieval methodology and assumptions made in the inversion are suitable for the smoke event investigated in this paper.

The anti-correlation between the retrieved ACAOD and COD observed for several CAR profiles is noted for COD mostly lesser than 10. This has been a known limitation of the color ratio method, in which the uncertainty in the retrieved ACAOD is estimated to be larger at lower COD and ACAOD values. This is because the retrieval domain at lower ACAOD/COD becomes narrower limiting the ability of algorithm, given several assumptions about aerosols and clouds, to accurately derive the aerosols and cloud fields.

Above discussion was added to the revised manuscript (Section 2.2).

General Comments: 1. In the introduction there should be mention, and comparison of a color ratio method for above cloud AOD by Meyer et al., 2015, that is applied to MODIS, and/or similarly from Peers et al., 2015. Additionally, there is little mention of the recent work based on the ORACLES measurements that follows from SAFARI. Potential to reference Redemann et al., 2020, and potentially LeBlanc et al., 2020.

Response: We added the suggested references: Meyer et al. 2015; Pistone et al. 2019; LeBlanc et al. 2020; Redemann et al. 2020.

2. Discussion of the impacts of the absorption properties of aerosol seems missing, particularly when referencing the color ratio technique in Section 2.2. Maybe a reference to the absorption properties from other radiative measurements during SAFARI; Bergstrom et al., 2003, or alternatively on the variations of the absorption as showcased by Pistone et al., 2019.

Response: The aerosol model used here in the ACAOD inversion is identical to the one employed in Jethva et al. [2016] paper, in which the MODIS retrievals of ACAOD were

found to be in very good agreement (RMSE~0.05 and 99% matchups within predicted uncertainty) against those directly measured from AATS sunphotometer. The results implied that the aerosol microphysical-optical properties assumed in the inversion that are essentially based on the long-term, ground-based AERONET inversion at an inland site Mongu, are suitable for ACAOD retrievals over the adjacent Atlantic Ocean.

Above discussion was added to the revised manuscript (Section 2.2)

3. Presentation of the figure 12, combining the AOD_cloudtop and AOD_sky might be better suited if there is inclusion of the measurement altitude, which might help indicate the partitioning. P.9 line 261: AOD from AATS would be representative either if directly above clouds, or below all significant layer of aerosol in the event of a clear-air-slot between cloud top and the bottom of the aerosol layer. It is suggested to add this caveat. The conclusion mentions this note again, but some care can be taken by careful data selection of sunphotometer data as presented by LeBlanc et al., 2020.

Response: The CAR BRDF measurements were obtained ~600 m above the clouds as pointed out in P.9 line 261 (or line 281 in the revised paper. So including the measurement altitude may not be necessary, plus it will make the plot more complex. The mean aircraft altitude is shown in Table 1 for each case. We would argue that the "clear-air-slot" concept is relative, where the concentration of aerosols in the slot is much lower than the layer above and/or below.

Specific Comments: 1. P.4 lines 121-122, AATS and 4STAR acronyms are not defined, please define and add pertinent citations.

Response: added: NASA Ames Airborne Tracking Sun Photometer (AATS) and Spectrometer for Sky-Scanning, Sun-Tracking Atmospheric Research (4STAR)

2. P.8 line 236: typo: 'betweent' should be 'between'

Response: the typo was corrected.

3. Table 1 shows an error value of 0.00 for much of the AATS AOD, this seems improbable and likely missing a significant digit. Additionally, there is no mention of what wavelength these AODs are reported (as compared to the retrieved ACAOD).

Response: We have corrected this anomaly based on actual errors derived from the AATS AOD. AOD are reported at wavelength= 0.500 $\mu$m as now indicated in Table 1.

4. Figure 4, There are no units on the colorbars, or the title is misleading – shouldn't it be radiance values in W/mȨ̈Ȩ2/nm/sr, or is it normalized radiances? If normalized radiance, it is normalized to what? The solar disc is apparently saturated, therefore if you normalize to that value, wouldn't that be misleading?

Response: The measured (sky or surface) radiance in any given direction is normalized by the solar irradiance incident on the top of the atmosphere, assuming mean Sun–Earth distance, and then converted to a non-dimensional quantity equivalent to effective BRF (or BRDF times $\pi$). This statement was added to Figure 4 caption.

5. Figure 4 a) & c), the solar disc seems to be not centered on the scattered light plot. The 0âÙȩ line does not seem to be in line with the principal plane.

Response: the appearance of the solar disc is not a reliable measure of asymmetry because of the saturation issue that we have. A plot of sky radiance as a function of azimuthal angle helps in identifying asymmetry due to errors in the geometrical correction. No action was taken.

6. Figure 8, the AOD above clouds retrieval at the solar disc seems drastically different than the surrounding region outside of the non-valid region.

Response: The spurious retrieval of AOD around Sun disk is a result of saturation in the CAR reflectance measurements and partly due to the inability of the RT model in simulating reflectance when directly looking at the Sun. This has been now clarified in the revised manuscript – Figure 8 caption.

7. Figure 12 – the figure caption lacks the identifier 'Figure 12:'

Response: We have added the identifier.

[Figure]

**Fig. 1.** Scattering phase function F11 of the carbonaceous aerosol model assumed in the aerosol inversion